# Photothermal recycling of waste polyolefin plastics into liquid fuels with high selectivity under solvent-free conditions

Yingxuan Miao[1,2], Yunxuan Zhao [1] ✉, Geoffrey I. N. Waterhouse [3], Run Shi [1], Li-Zhu Wu [1] & Tierui Zhang [1,2] ✉

The widespread use of polyolefin plastics in modern societies generates huge amounts of plastic waste. With a view toward sustainability, researchers are now seeking novel and low-cost strategies for recycling and valorizing polyolefin plastics. Herein, we report the successful development of a photothermal catalytic recycling system for transforming polyolefin plastics into liquid/waxy fuels under concentrated sunlight or xenon lamp irradiation. Photothermal heating of a Ru/TiO$_2$ catalyst to 200–300 °C in the presence of polyolefin plastics results in intimate catalyst-plastic contact and controllable hydrogenolysis of C-C and C-H bonds in the polymer chains (mediated by Ru sites). By optimizing the reaction temperature and pressure, the complete conversion of waste polyolefins into valuable liquid fuels (86% gasoline- and diesel-range hydrocarbons, C$_5$-C$_{21}$) is possible in short periods (3 h). This work demonstrates a simple and efficient strategy for recycling waste polyolefin plastics using abundant solar energy.

Over 8 billion tons of plastics have been produced to date, with ~80% of this production ending up in landfills or as pollutants in natural environments. This is both wasteful and an ecological disaster (e.g., the accumulation of microplastics in the rivers, oceans, and soils)[1,2]. Polyolefin plastics, including low-density polyethylene (LDPE), high-density polyethylene (HDPE), and polypropylene (PP), are indispensable plastics in our modern lives, accounting for ~57% of global plastic production[1,3,4]. Abundant and inexpensive olefin feedstocks from the petrochemical industry enable the low-cost production of polyolefin plastics for single-use or short-term-use applications, such as packaging and disposable medical devices[5,6]. Currently, ~220 million tons of polyolefin plastics are generated annually, with production expected to quadruple by 2050. Global recycling rates are only 5% for LDPE, 10% for HDPE, and <1% for PP[1,7]. Estimates suggest that if all polyolefin plastics could be efficiently recycled, the value of the extracted products would exceed US$100 billion each year[8]. Further, entirely new industries could be built around recycling discarded polymers[7]. However, traditional approaches used in polyolefin plastic recycling are energy intensive due to the fact that the C-C and C-H bonds in polyolefins are very inert[9,10]. Pyrolysis-based recycling routes require harsh operating conditions (temperatures > 500 °C), demanding vast energy input and greatly increasing the cost of polyolefin plastic recycling[8,10]. Advanced hydrogenolysis techniques allow plastic recycling at lower reaction temperatures (~300 °C). However, external heat still needs to be supplied to drive the process[5,6,8,11]. Harnessing solar energy to drive polyolefin plastic recycling is a possible approach for minimizing energy inputs[9]. For instance, many plastics can be slowly mineralized by photocatalytic degradation over TiO$_2$[12], ZnO[13], and NiAl$_2$O$_4$ spinels[14] under ultraviolet (UV) irradiation or visible (Vis) irradiation. As an example, Reisner et al. showed that alkaline hydrolysis of polyethylene terephthalate, polylactic acid, and polyurethane could yield monomers[9], which in the presence of a photocatalyst could be oxidized to produce various organic chemicals (e.g., carboxylic acids and aldehydes) with concomitant H$_2$ evolution. Xie's group reported that polyethylene, polypropylene, and polyvinyl chloride can be oxidized to CO$_2$, then photocatalytically reduced to acetic acid by single-unit-

[1]Key Laboratory of Photochemical Conversion and Optoelectronic Materials, Technical Institute of Physics and Chemistry, Chinese Academy of Sciences, 100190 Beijing, China. [2]Center of Materials Science and Optoelectronics Engineering, University of Chinese Academy of Sciences, 100049 Beijing, China. [3]School of Chemical Sciences, The University of Auckland, Auckland 1142, New Zealand. ✉e-mail: yunxuan@mail.ipc.ac.cn; tierui@mail.ipc.ac.cn

cell thick $Nb_2O_5$ layers[15]. Whilst these works show promise, rates of photocatalytic processes are too low to warrant serious interest from plastics recyclers[9,10]. Alternative approaches need to be found for solar-driven polyolefin plastic recycling.

Recently, photothermal catalysis has attracted a lot of interest in processes such as Fischer-Tropsch synthesis[16], $CO_2$ conversion[17,18], $NH_3$ synthesis[19], and the degradation of pollutants[20,21]. In photothermal catalysis, a catalyst absorbs light strongly over the UV-Vis-near infrared (NIR) region, leading to rapid local heating of the catalyst to temperature ranges where thermal catalytic reactions become possible[22]. Photothermal catalysis often combines the benefits of traditional thermal catalysis and photocatalysis, with synergies involving these two types of the catalytic process often boosting performance[18,22–24]. In the case of photothermal recycling of polyolefin plastics, reaction temperatures above the melting point of the plastics (melting temperatures by differential scanning calorimetry (DSC)[25]: 112 °C for LDPE, 138 °C for HDPE, 169 °C for PP) will be beneficial for (1) increasing the mobility of polymeric chains and (2) increasing the contact between the polymers and the photothermal catalyst, thereby improving the probability of C-C or C-H scission processes in the presence of a suitable catalyst[26]. Further, UV irradiation of polyolefins can activate inert polymeric chains and generate active radical species (such as the allylic radicals in the irradiated polyethylene), thereby offering pathways for polymer conversion into other products[27–29]. Based on this information, we hypothesized that photothermal recycling of polyolefin plastic wastes into valuable liquid fuels should be possible in the presence of a suitable catalyst with good activity for C-C and C-H scission reactions.

Herein, we report a photothermal route for transforming waste polyolefin plastics into value-added products using concentrated sunlight or a xenon (Xe) lamp to drive the chemical transformations. The Ru/TiO$_2$ catalyst used in this work contains small Ru nanoparticles uniformly dispersed over a P25 TiO$_2$ support. The ruthenium (Ru) nanoparticles are heated rapidly to several hundred degrees Celsius under UV-Vis-NIR irradiation (leading to polymer melting), whilst also acting as active sites for C-C and C-H bond scission in the polymer chains. By varying the photothermal reaction temperature and pressure, different products can be produced, including high-purity methane and liquid/waxy fuels, or valuable liquid fuels (gasoline- and diesel-range hydrocarbons, $C_5$-$C_{21}$) in high yields. The photothermal polyolefin recycling system works efficiently for a range of polyolefin feedstocks (e.g., LDPE, HDPE, ultrahigh molecular weight polyethylene (UHMWPE), and PP) and commercial LDPE bags. Results encourage the use of photothermal catalytic technologies in the recycling to polyolefin plastics.

## Results
### Construction of the photothermal polyolefin plastic recycling system

We aimed to develop a solvent-free photothermal method to recycle various types of polyolefin plastic. Our approach utilized a Ru/TiO$_2$ catalyst, taking advantage of the excellent C-C and C-H activation ability of Ru nanoparticles and the high thermal and chemical stability of TiO$_2$ as a support[30–32]. Details about the synthesis of the Ru/TiO$_2$ catalyst are provided in the "Methods" section. As shown in Supplementary Fig. 1a, the dark gray Ru/TiO$_2$ catalyst exhibited strong light absorption across the ultraviolet, visible, and near-infrared regions, thus offering full-spectrum sunlight utilization for photothermal heating. Powder X-ray diffraction (XRD) data for the Ru-TiO$_2$ catalyst showed only peaks due to the P25 TiO$_2$ support (i.e., anatase and rutile in 6:1 weight ratio, Supplementary Fig. 1b). No peaks due to Ru-containing species were seen by XRD, implying that any Ru nanoparticles were likely very small. Transmission electron microscopy (TEM) images and energy dispersive X-ray (EDX) element maps showed the Ru/TiO$_2$ catalyst that contains Ru nanoparticles (size ~2.5 nm) uniformly dispersed on the TiO$_2$ support (Supplementary Fig. 2a–c).

An intimate catalyst-plastic contact is requisite for high-performance recycling. In traditional photocatalytic LDPE recycling, the solid catalyst and solid LDPE exists as separate entities and make

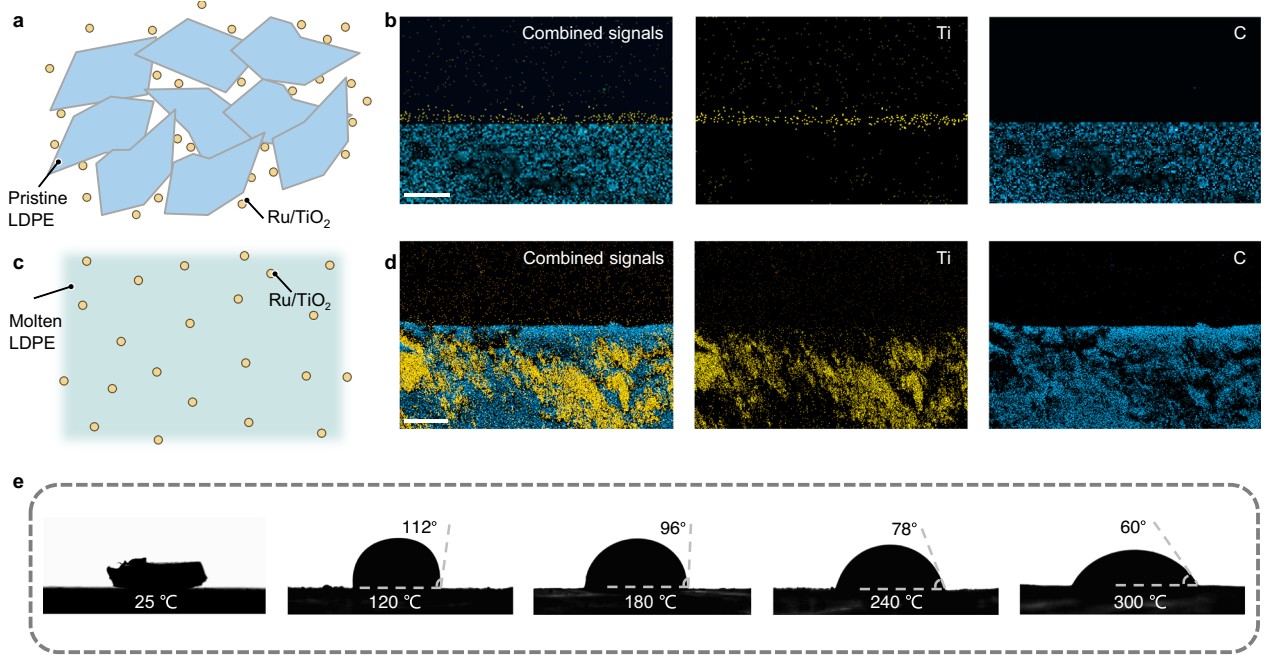

**Fig. 1 | Contact between the Ru/TiO$_2$ catalyst and LDPE in photothermal recycling system.** Schematic illustration of contact between catalyst and LDPE in **a**, traditional photocatalytic recycling and **c**, photothermal recycling. **b** EDX maps of cross-sectional SEM image for a physical mixture of a LDPE granule and Ru/TiO$_2$ sample (state likes traditional photocatalytic LDPE recycling). Scale bar, 100 μm. **d** EDX maps of cross-sectional SEM image for a quenched molten LDPE-Ru/TiO$_2$ sample (state likes photothermal LDPE recycling). Scale bar, 5 μm. **e** Pristine LDPE and molten LDPE droplet contact angles on a glass substrate coated with Ru/TiO$_2$ at various temperatures in an argon (Ar) atmosphere.

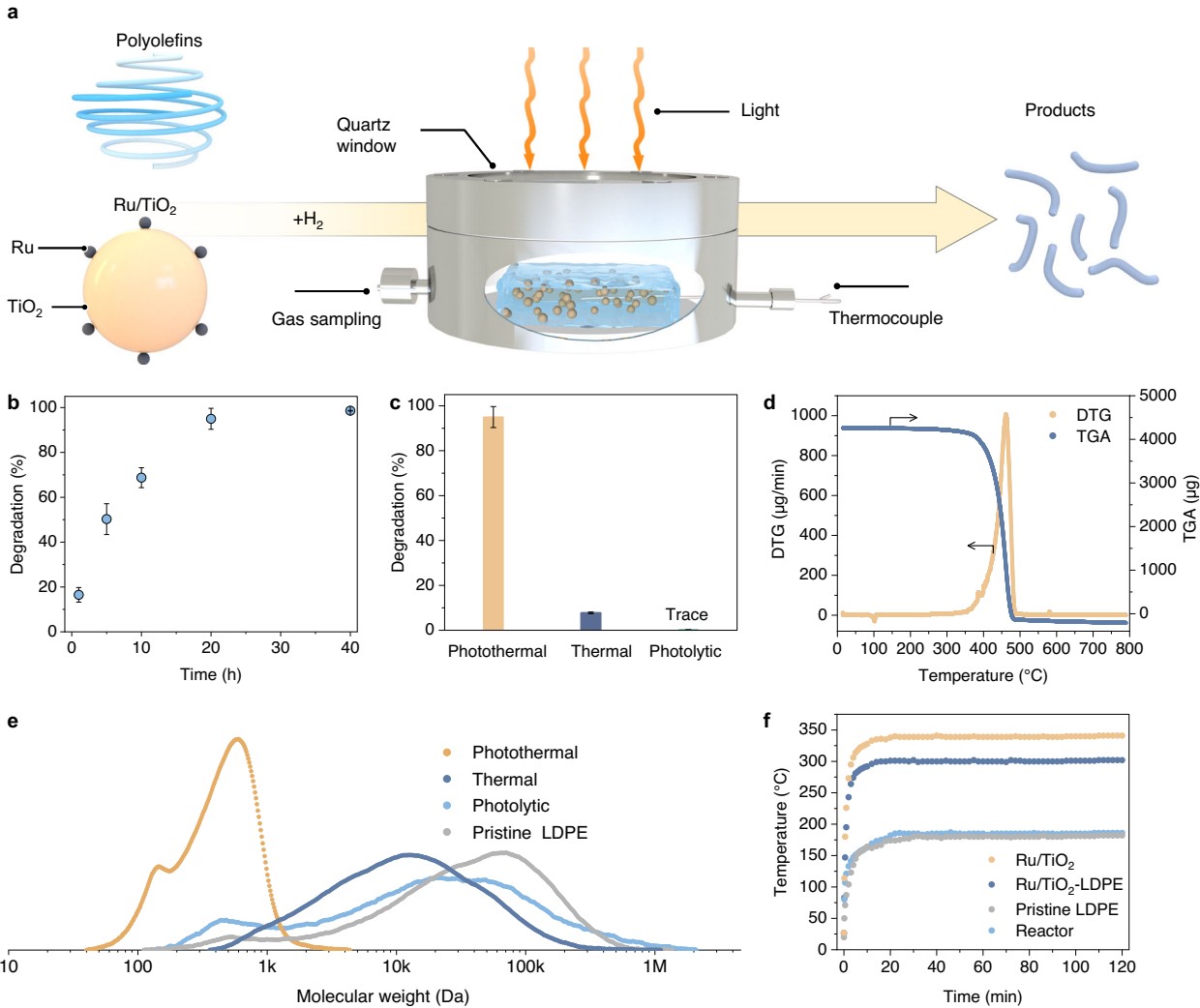

**Fig. 2 | Photothermal degradation of LDPE. a** Schematic illustration of the photothermal polyolefin plastic recycling system. The thermocouple is used for temperature detection. The gaseous products are sampled through a gas sampling port and liquid/waxy products are collected and analyzed post-reaction by GPC, high-temperature gas chromatography (HTGC), and $^1H$ nuclear magnetic resonance ($^1H$ NMR). **b** Degradation percentages of LDPE as a function of photothermal reaction time over Ru/TiO$_2$ at 300 °C under 1 bar H$_2$/Ar (v/v = 30/70). **c** Degradation percentages of LDPE under various reaction conditions for 20 h (i.e., photothermal degradation at 300 °C, thermal degradation at 300 °C, and photolysis at ambient temperature). **d** Thermogravimetric analysis (TGA) and differential thermal gravimetry (DTG) of pristine LDPE under an Ar atmosphere. **e** GPC analysis of molecular weight ($M_{w-PD}$ = 0.49 kDa, $M_{w-TD}$ = 25.0 kDa, $M_{w-PL}$ = 66.5 kDa) and dispersity ($Đ_{PD}$ = 1.6, $Đ_{TD}$ = 4.9, $Đ_{PL}$ = 23.5) of corresponding samples. The vertical axis (d$w$/dlog$M$) was omitted for clarity. **f** Temperature profiles of the ambient-pressure reactor, LDPE, Ru/TiO$_2$, and a mixture of Ru/TiO$_2$ and LDPE (1:4 by weight) under illumination from a Xe lamp at the same light intensity (3.0 W cm$^{-2}$).

poor contact (Fig. 1a, b). On the other hand, an intimate catalyst-plastic contact was realized in the photothermal recycling process since light from a Xe lamp or concentrated sunlight resulted in local heating of Ru/TiO$_2$ and melting of the solid LDPE into molten LDPE (Fig. 1c). The EDX maps of cross-sectional scanning electron microscopy (SEM) for a quenched molten LDPE-Ru/TiO$_2$ sample in Fig. 1d verified the intimate contact between the Ru/TiO$_2$ catalyst and the LDPE. Figure 1e showed that the contact angles between a molten LDPE droplet and the Ru/TiO$_2$ powder (on a glass substrate) decreased with increasing temperature in the range of 120–300 °C (from contact angle = 112° at 120 °C to 60° at 300 °C). The measured contact angle at 300 °C further decreased under Xe lamp or UV irradiation (Supplementary Fig. 3). Hence, LDPE wetted Ru/TiO$_2$ better at higher temperatures or under light irradiation, resulting in superior catalyst-plastic contact under photothermal recycling conditions compared with traditional photocatalytic recycling[33].

Figure 2a shows the photothermal catalytic polyolefin plastic recycling system. Typically, a mixture of Ru/TiO$_2$ catalyst and LDPE was

loaded into the reactor. Light from a Xe lamp (Supplementary Fig. 4) or concentrated sunlight was then directed through the quartz window of the photothermal reactor, resulting in Ru/TiO$_2$ heating and melting of the LDPE to form a viscous dispersion of catalyst in molten LDPE. The reaction temperature shows a linear correlation with the light intensity of the Xe lamp (Supplementary Fig. 5). The catalytic activity of Ru/TiO$_2$ caused scission of C–C and C–H bonds in LDPE with the aid of H$_2$. By this approach, LDPE and other polyolefin plastics were completely degraded and converted into value-added products. In preliminary polyolefin plastic degradation experiments, LDPE (80 mg, $M_w$ = 68.7 kDa, Đ = 11.4) was mixed with Ru/TiO$_2$ (20 mg, containing 2.0 wt.% Ru) and the resulting mixture was transferred to the photothermal reactor (Supplementary Fig. 6) and photothermally heated at temperatures between 200–350 °C in a 1 bar H$_2$/Ar (v/v = 30/70) gas mixture. The LDPE degradation percentage (labeled as $R_d$) gradually increased with photothermal reaction temperature up to 300 °C (light intensity = 3.0 W cm$^{-2}$) and also reaction time, approaching nearly 100% after 20 h at 300 °C (Fig. 2b and Supplementary Fig. 7). The

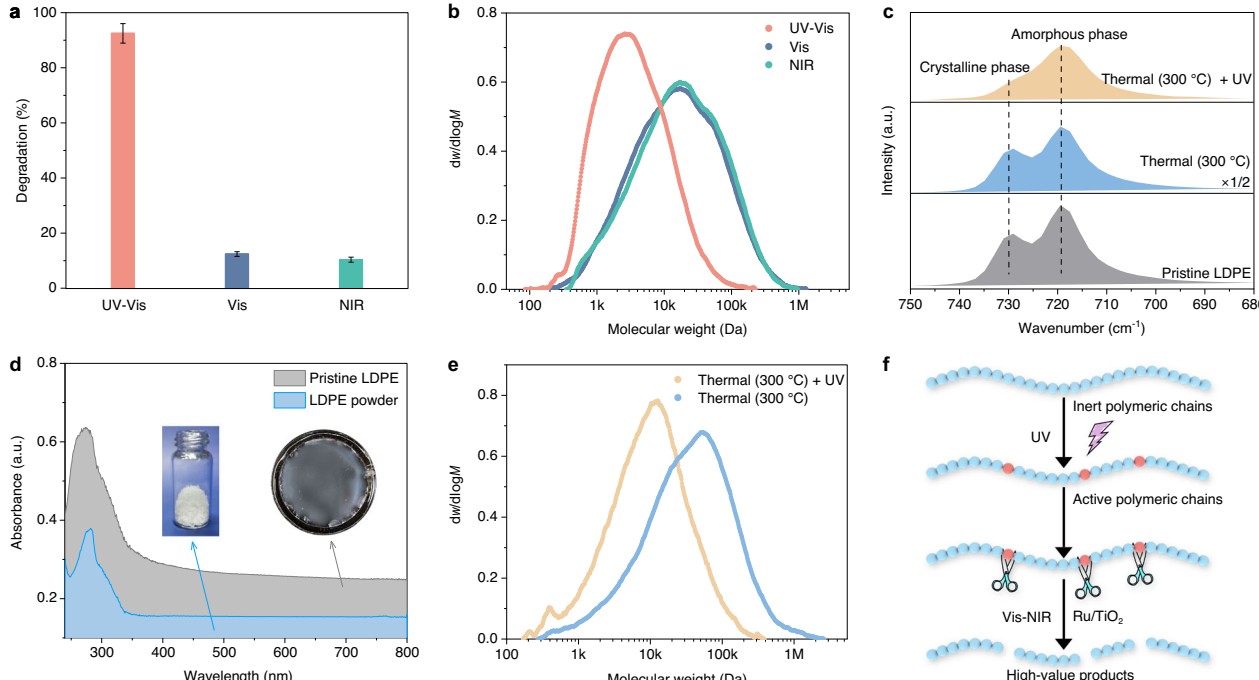

**Fig. 3 | Proposed reaction mechanism. a** Degradation of LDPE over Ru/TiO$_2$ under UV-Vis, Vis, and NIR irradiation at 300 °C under 1 bar H$_2$/Ar (v/v = 30/70) after 20 h reaction. **b** GPC analysis and distribution of polymeric residues after degradation over Ru/TiO$_2$ under UV-Vis, Vis, and NIR irradiation at 300 °C under 1 bar H$_2$/Ar (v/v = 30/70) after 5 h reaction (UV-Vis: $M_w$ = 6.8 kDa and $Đ$ = 3.6; Vis: $M_w$ = 40.4 kDa and $Đ$ = 7.4; NIR: $M_w$ = 41.9 kDa and $Đ$ = 7.0). **c** ATR-IR spectra of pristine LDPE, Thermal (pristine LDPE at 300 °C under 1 bar H$_2$/Ar (v/v = 30/70) after 5 h reaction

without any catalyst), and Thermal + UV (same condition of Thermal with extra UV light). **d** UV-Vis diffuse reflectance spectrum of LDPE powder and pristine LDPE under ambient temperature. **e** GPC distributions of LDPE residue after 5 h reaction (300 °C and 1 bar H$_2$/Ar (v/v = 30/70)) without any catalyst (Thermal: $M_w$ = 80.2 kDa, $Đ$ = 7.7; Thermal + UV: $M_w$ = 19.1 kDa, $Đ$ = 5.0). **f** Schematic illustration of the proposed photothermal reaction mechanism.

weight-average molecular weight ($M_w$) of LDPE determined by gel permeation chromatography (GPC) dropped from 68.7 kDa (for pristine LDPE) to 12.9 kDa after 1 h at 300 °C (Supplementary Fig. 8). On extending the reaction time to 10 h, the $M_w$ further dropped to 3.4 kDa, implying the efficient scission of C-C bonds in the LDPE polymeric chains. Random scission of C-C bonds and insufficient cracking of the high-molecular-weight portion of polymer residues during thermal degradation of LDPE over Ru/TiO$_2$ led to wide dispersity ($Đ = M_w/M_n$; Supplementary Fig. 9)[6,8]. For the photothermal degradation of LDPE in the presence of Ru/TiO$_2$, the dispersity is narrowed and progressively reduced, indicating efficient cracking of large polymer molecules. Under photothermal conditions, 95.0% of the LDPE was degraded at 300 °C after 20 h reaction (Fig. 2c, d). Conversely, minor LDPE degradation occurred under direct thermal heating in the dark at 300 °C ($R_d$ = 7.8%) or under direct photolysis at room temperature (negligible degradation). As shown in Fig. 2e, the polymeric residue of the thermal degradation (TD) and photolysis (PL) experiments had a large molecular weight ($M_{w\text{-}TD}$ = 25.0 kDa and $M_{w\text{-}PL}$ = 66.5 kDa, respectively) and wide dispersity ($Đ_{TD}$ = 4.9 and $Đ_{PL}$ = 23.5, respectively). For the photothermal degradation (PD) experiments after 20 h at 300 °C, liquid/waxy products centered around C$_{34}$ were obtained rather than polymeric residues, resulting in a $M_{w\text{-}PD}$ of 490 Da (i.e., 0.49 kDa) and a $Đ_{PD}$ of 1.6 (specific products formed are discussed below). Intriguingly, the $Đ_{PL}$ and $M_{w\text{-}PL}$ values of the polymeric residues of the photolysis experiments were wider and smaller, respectively, than the corresponding values for pristine LDPE ($Đ_{PL}$ = 23.5 and $M_{w\text{-}PL}$ = 66.5 kDa for photolysis process, $Đ$ = 11.4 and $M_w$ = 68.7 kDa for pristine LDPE). The larger $Đ$ value for the polymer residue of the photolysis experiments, compared to pristine LDPE, may have been due to UV-promoted crosslinking, while the decreased $M_w$ was probably due to the photodegradation of LDPE. Results suggested that cross-linking and scission

of polymer chains were competitive processes during the photolysis of LDPE[34–37]. For the photothermal degradation experiments, both the $Đ_{PL}$ and $M_w$ values were much lower than those of pristine LDPE, highlighting the advantages of the photothermal degradation process. In the absence of the Ru/TiO$_2$ catalyst, a temperature of only 180 °C can be reached at 3.0 W cm$^{-2}$, with negligible photothermal degradation of LDPE occurring ($R_d$ < 1%, $M_w$ = 56.0 kDa at 180 °C utilizing direct irradiation from a Xe lamp) (Fig. 2f and Supplementary Fig. 10). Results demonstrated that the Ru/TiO$_2$ catalyst was essential for photo-induced heating and efficient photothermal catalytic degradation of LDPE. The Ru/TiO$_2$ catalyst was stable under the photothermal testing conditions used in the current work, with no phase or structure changes found following four cycles of LDPE degradation experiments at 300 °C (Supplementary Figs. 11–13).

In conventional thermal catalytic processes, the entire reactor is heated externally and then reactants are uniformly heated through diffusive heat transfer and thermal convection[38]. In the case of photothermal catalytic processes, solar radiation causes local catalyst heating, thereby minimizing the required energy input[38–40]. In the Ru/TiO$_2$ system, various nonradiative processes are responsible for the local catalyst heating. Figure 2f shows that under Xe lamp irradiation, the catalyst temperature increased rapidly and stabilized after ~10 min, with the temperature controlled by the light intensity from the Xe lamp. The photothermal temperature profile for the Ru/TiO$_2$ and LDPE mixture was similar to that of the Ru/TiO$_2$ catalyst, with a temperature of 300 °C easily achieved. Local heating at such temperatures was expected to promote the catalytic degradation of LDPE[18,38,41]. As shown in Supplementary Fig. 14, the degradation of LDPE under photothermal local heating conditions was enhanced compared to that under thermal heating conditions at each temperature studied.

**Table 1 | Comparison of photothermal or thermal recycling of various polyolefin plastics over Ru/TiO$_2$**

| Types of plastics | $M_w$ (kDa) | Temp. (°C) | Time (h) | $R_d$ (%) | Selectivity (%) | |
|---|---|---|---|---|---|---|
| | | | | | Gas (C$_1$-C$_4$) | Liquid/ wax (C$_{5+}$) |
| PD-LDPE | 68.7 | 300 | 20 | 95.0 | 9 | 91 |
| TD-LDPE | 68.7 | 300 | 20 | 7.8 | 11 | 89 |
| PD-UHMWPE | 3k–6k$^a$ | 300 | 40 | 90.0 | 5 | 95 |
| TD-UHMWPE | 3k–6k$^a$ | 300 | 40 | 3.3 | 15 | 85 |
| PD-HDPE | 79.3 | 300 | 20 | 87.8 | 3 | 97 |
| TD-HDPE | 79.3 | 300 | 20 | 6.5 | 7 | 93 |
| PD-PP | 304.2 | 300 | 20 | 93.9 | 5 | 95 |
| TD-PP | 304.2 | 300 | 20 | 2.3 | 17 | 83 |
| PD-LDPE bags | 110.7 | 300 | 20 | 97.3 | 3 | 97 |
| TD-LDPE bags | 110.7 | 300 | 20 | 4.1 | 9 | 91 |

$^a$The $M_w$ was determined by GPC or provided by the producer.
Reaction conditions: 300 °C, 1 bar H$_2$/Ar (v/v = 30/70), 80 mg polyolefin, 20 mg Ru/TiO$_2$. PD represents photothermal recycling, TD represents thermal recycling, and $R_d$ represents the degradation percentage.

**Proposed photothermal reaction mechanism**

Whilst local heating is able to promote polyolefin plastic degradation, other factors potentially contributed to the degradation of LDPE under photothermal recycling conditions. To study the mechanism of LDPE degradation under photothermal heating conditions, we explored LDPE degradation under different light illumination regimes (i.e., UV-Vis, Vis, and NIR irradiation) in the presence of Ru/TiO$_2$. Photothermal LDPE degradation at 300 °C with Ru/TiO$_2$ after 20 h under visible light ($R_d$ = 12.4%) and NIR light ($R_d$ = 10.4%) were similar to that under direct thermal degradation conditions ($R_d$ = 7.8%), indicating that the primary contribution of Vis and NIR irradiation in the photothermal recycling system was local heating (Fig. 3a). Under UV-Vis irradiation at 300 °C, the LDPE degradation percentage increased to 92.5%, similar to photo-thermal degradation performance under UV-Vis-NIR conditions at the same reaction temperature (95%). The GPC distributions of the polymer residues showed that the type of illumination (i.e., UV-Vis, Vis, and NIR light) strongly influenced the molecular weight of the remaining LDPE residues (Fig. 3b). The presence of UV light clearly decreased the average molecular weight of the polymer residues. A possible promotion effect due to local surface plasmon resonance (LSPR) of the Ru nanoparticle can largely be discounted on the basis of finite-difference time-domain simulations (Supplementary Fig. 15) and catalytic tests. The similar GPC distributions of the polymeric residues from thermal degradation experiments at 300 °C and thermal degradation experiments at 300 °C + 365 nm irradiation over Ru/TiO$_2$ eliminate the possibility of a significant photocatalytic effect caused by TiO$_2$ or a LSPR effect caused by the supported Ru nanoparticles (Supplementary Fig. 16). Hence, the UV irradiation ($\lambda$ < 365 nm) from the Xe lamp or sunlight probably directly acted on LDPE and promoted degradation. Evidence for this was seen in experiments without any catalyst (Supplementary Fig. 17). Under Xe lamp irradiation, photothermal LDPE degradation occurred to a greater extent (10.0% degradation) than that under thermal degradation conditions (1.3% degradation).

To further explore the effect of the UV irradiation on LDPE degradation without any catalyst, attenuated total reflectance infrared spectroscopy (ATR-IR) was used to probe the crystalline phase (730 cm$^{-1}$) and amorphous phase (720 cm$^{-1}$), both of which typically coexist in LDPE[42,43]. Crystalline domains are often considered more inert during LDPE degradation than amorphous domains[44–46]. For pristine LDPE and the polymeric residues after thermal degradation at 300 °C, peaks at 730 cm$^{-1}$ and 720 cm$^{-1}$ were observed, while after thermal + UV treatment at 300 °C, the 730 cm$^{-1}$ peak had almost disappeared (Fig. 3c). This indicates that thermal + UV treatment can destroy the crystalline domains of LDPE with the remaining polymer chains having been modified substantially. Furthermore, UV light is able to cause scission of the C-C bonds on account of the comparatively low bond energy of C-C in polymers (~3.44 eV)[29]. As shown in Fig. 3d, both the LDPE powder and the pristine LDPE displayed a strong absorption below 350 nm (~3.54 eV). Absorption of UV light with the aid of thermal treatment at 300 °C caused a decrease in LDPE molecular weight ($M_w$ = 19.1 kDa versus $M_w$ = 80.2 kDa), which is a direct result of chain scission reactions (Fig. 3e)[47]. Thermal decomposition measurements of LDPE without any catalyst indicated the preferential scission of terminal C-C bonds, whereas under UV irradiation the scission preferentially appears at internal C-C bonds (Supplementary Fig. 18). In summary, the chemical inertness of LDPE was lessened by the presence of UV irradiation ($\lambda$ < 365 nm) during the photothermal degradation reaction.

Intriguingly, whilst UV light was shown to lessen the chemical inertness of LDPE, the LDPE residues from the photothermal or thermal reactions with Ru/TiO$_2$ showed similar melting temperatures, as probed by DSC (Supplementary Fig. 19). This implied a similar reaction path. This prompted us to perform control experiments under thermal conditions to estimate the extent of thermocatalytic degradation of LDPE over Ru/TiO$_2$ and some reference catalysts. Negligible LDPE degradation occurred with heating at 300 °C in the absence of catalyst (1.3% over 20 h). A 6-fold improvement in LDPE thermal degradation (up to 7.8%) was achieved in the presence of Ru/TiO$_2$, together with a decrease in the polymer $M_w$, indicating the scission of C-C bonds (Supplementary Fig. 20). Similar results have been reported for the hydrogenolysis of polypropylene[32], and light alkanes that involved in a series of dehydrogenation steps followed by hydrogenation over Ru nanoparticles[48,49]. As expected, Ru powder showed better thermal LDPE degradation performance than pure TiO$_2$ (Supplementary Fig. 20). To verify our conclusion, we carried out low-temperature hydrogenolysis experiments using n-hexadecane as a model compound based on recent benchmarking experiments of Dyson et al.[50]. The conversion of n-hexadecane on the Ru powder was ~25 times higher than on TiO$_2$, confirming that Ru nanoparticles were the main active sites for C-C bonds scission and hydrogenolysis was the main reaction route (Supplementary Figs. 21–23).

Based on the findings above, we proposed a mechanism for the photothermal degradation of LDPE over the Ru-TiO$_2$ catalyst (Fig. 3f). Photothermal heating of the Ru/TiO$_2$ catalyst under full-spectrum Xe lamp irradiation (or concentrated sunlight) causes local photothermal heating of the catalyst and melting of the polymer. UV light ($\lambda$ < 365 nm) activates the LDPE chains, creating reaction sites for scission by Ru nanoparticles on the Ru/TiO$_2$ catalyst. The mechanism accounts for the efficient photothermal recycling process of LDPE into small gaseous and liquid/waxy hydrocarbon products through the synergistic utilization of UV, Vis, and NIR light.

**Development of an efficient photothermal polyolefin plastic recycling system**

Following the mechanistic study, we further applied the photothermal recycling system to other types of polyolefin plastics (Table 1). UHMWPE ($M_w$ = 3000–6000 kDa, data from producer), HDPE ($M_w$ = 79.3 kDa, Đ = 3.5), and PP ($M_w$ = 304.2 kDa, Đ = 2.4) could be effectively degraded in 20 h at 300 °C and 1 bar H$_2$/Ar (v/v = 30/70) gas mixture in the presence of the Ru/TiO$_2$ catalyst ($R_d$ = 90.0% for UHMWPE, 87.8% for HDPE, and 93.9% for PP). To further test the universality of the photothermal recycling system, commercial LDPE bags ($M_w$ = 110.7 kDa, Đ = 3.3) containing various additives were also selected for photothermal recycling. A degradation percentage of 97.3%

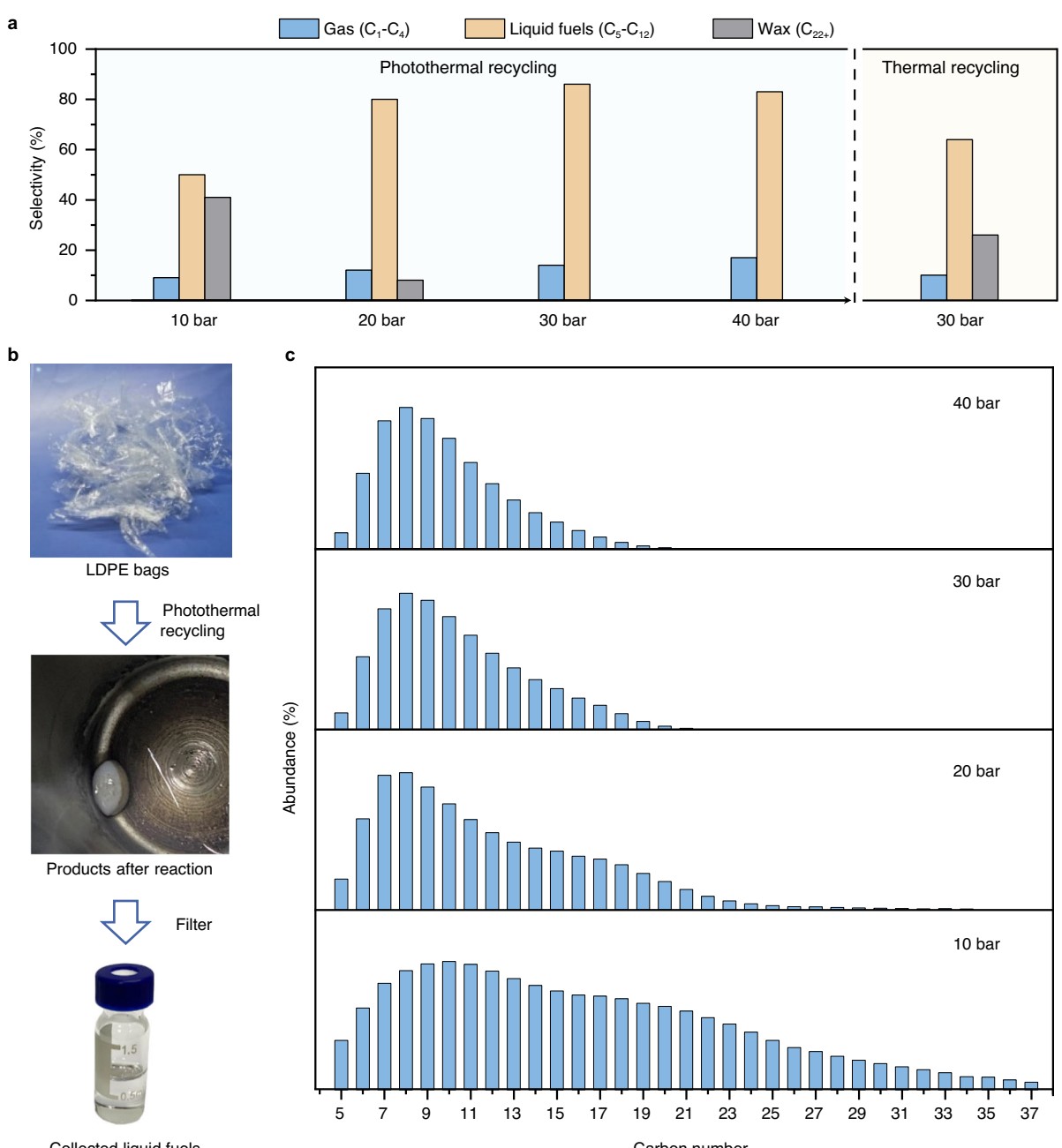

**Fig. 4 | Pressure-dependent product distributions of the photothermal polyolefin plastic recycling system.** **a** Product selectivities of the high-pressure (220 °C for 3 h over Ru/TiO$_2$ catalyst) photothermal recycling of LDPE bags. **b** Process of high-pressure photothermal recycling of LDPE bags under 30 bar H$_2$/ N$_2$ (v/v = 70/30). **c** Distributions of liquid products obtained from the photothermal recycling of LDPE bags at 220 °C for 3 h at different reaction pressures (10, 20, 30, 40 bar H$_2$/N$_2$ (v/v = 70/30)) over Ru/TiO$_2$.

was realized in 20 h (300 °C and 1 bar H$_2$/Ar (v/v = 30/70)), confirming the viability of the method. By comparison, attempts to degrade UHMWPE, HDPE, PP, or the LDPE bags by thermal catalytic methods were largely unsuccessful ($R_d$ = <8.0% for all these polyolefin plastics), highlighting the merits of the photothermal recycling system. For both the photothermal and thermal recycling systems, the amounts of generated gaseous products (3–17% of all products) for each type of polyolefin plastic were relatively low compared with the liquid/waxy products (91–97% of all products for the photothermal recycling system, 83–93% for thermal recycling system).

The gaseous and liquid/waxy products generated in the photothermal recycling system were studied in detail (for photothermal LDPE recycling). As the reaction time increased in the presence of the Ru/TiO$_2$ catalyst at 300 °C and 1 bar H$_2$/Ar (v/v = 30/70), the selectivity to CH$_4$ increased and approached 100% after 40 h reaction (this selectivity is purely based on the gaseous products, not the total products which included liquids/waxes; Supplementary Fig. 24). According to Supplementary Fig. 25, CH$_4$ primarily originates from the hydrogenolysis of light hydrocarbons. The selectivity of gaseous products over a single component (e.g., Ru nanoparticles or TiO$_2$) verified that Ru sites played a vital role in methanation (Supplementary Fig. 26). The high degree of methanation relates to the direct terminal C-C cleavage or/and the surface cascade of consecutive C-C cleavage reactions over the surface of metallic Ru at low hydrogen partial pressure[51,52]. Under the same reaction conditions, >90% CH$_4$ selectivity in gaseous products when repeatedly recycling LDPE or using other

types of polyolefin plastics (Table 1 and Supplementary Figs. 27–29). The liquid/waxy products for photothermal LDPE recycling showed a $C_{27}$-centered distribution (Supplementary Fig. 30a), which was close to the results of $M_{w-PD}$ determination (490 Da). The collected products were analyzed by $^1$H NMR spectroscopy, with the peaks in the region of 0–2 ppm being due to -$CH_3$, -$CH_2$, and -CH signals of alkane chains (Supplementary Fig. 30b)[8]. Similar carbon number distributions were obtained for the liquid/waxy products formed during photothermal recycling of the other polyolefin plastics or repeated cycles (Supplementary Figs. 31–35), suggesting the good repeatability and universality of the photothermal catalytic recycling system.

In order to further explore the advantages of the photothermal recycling system, we performed photothermal LDPE bags recycling experiments at elevated reaction pressures utilizing a high-pressure photothermal stainless reactor (Supplementary Fig. 36). Initial experiments were performed at 180, 200, and 220 °C and 30 bar $H_2/N_2$ (v/v = 70/30). The photothermal recycling system outperformed the thermal recycling system at all temperatures (Supplementary Fig. 37). On raising the reaction temperature to 220 °C, the LDPE bags was completely degraded within 3 h under the photothermal recycling conditions. In the photothermal recycling experiments, a substantial amount of waxy products were obtained at a pressure of 10 bar (Fig. 4a). The selectivity of waxy products decreased and liquid fuels gradually increased as the pressure elevated. Remarkably, 100% of liquid products at a pressure of 30 bar were located within the gasoline and diesel range ($C_5$-$C_{21}$) during photothermal recycling, representing 86% of the total products, whereas only 64% of the products located in this range for thermal recycling (Fig. 4b and Supplementary Table 1). By prolonging the reaction time for thermal recycling from 3 h to 9 h, the selectivity to liquid products approached that achieved during photothermal recycling over 3 h, although 7% wax still remained (Supplementary Fig. 38 and Supplementary Table 2). On further prolonging the thermal reaction time to 12 h, the selectivity of liquid fuels steeply declined with additional gas products being formed. Under the photothermal recycling conditions, the distribution of the products shifted towards lower carbon numbers as the reaction pressure increased in the range of 10 bar to 30 bar, remaining almost unchanged with the increase of the pressure to 40 bar (Fig. 4c and Supplementary Figs. 39–43). Poor hydrogenolysis caused by a deficiency of chemisorbed hydrogen atoms under low hydrogen partial pressure leads to wax production[49]. Hence, low hydrogen partial pressures (e.g., below 10 bar) yield substantial amounts of wax products, whilst sufficiently high hydrogen partial pressures (e.g., 20 to 30 bar) results in a high selectivity to liquid fuels. The photothermal recycling method also offered significant advantages in the recycling of isotactic polypropylene, implying its good universality (Supplementary Fig. 44). In order to explore the feasibility of photothermal polyolefins recycling on a large scale, we carried out up-scaled photothermal recycling at a 5 g scale (Supplementary Fig. 45). 87% selectivity of gasoline- and diesel-range hydrocarbons ($C_5$-$C_{21}$) was achieved, verifying the scalability of the photothermal recycling method.

Furthermore, we tested the performance of photothermal polyolefin plastic recycling system using concentrated sunlight (Supplementary Fig. 46a). The concentrated sunlight rapidly heated the catalyst, resulting in reaction temperatures of 200 ± 20 °C, 300 ± 20 °C, or 400 ± 20 °C depending on the concentrated light intensity (Supplementary Fig. 46b). The LDPE degradation efficiencies under concentrated sunlight were impressive, and similar to those results obtained under Xe lamp irradiation (Supplementary Fig. 46c). In addition, a simple technoeconomic analysis of industrial polyethylene hydrogenolysis over Ru/TiO$_2$ was performed using Aspen Plus simulation software. The analysis revealed that the reactor consumed most of the energy (347.9 kW/h), accounting for 90.0% of the whole process (Supplementary Fig. 47 and Supplementary Table 3).

Hence, massive cost savings in polyolefin plastic recycling should be possible through harnessing concentrated sunlight[53]. Hence, we envisage that polyolefin plastics will become a future feedstock for the chemical and energy sectors, rather than ending up in landfills. By valorizing plastic wastes utilizing the solar-driven recycling routes, together with phasing out single-use plastic items (especially food packaging), current environmental issues linked to plastic wastes may be avoided.

## Discussion

We constructed a photothermal catalytic system for the recycling of polyolefin plastics (LDPE, HDPE, UHMWPE, PP, and commercial LDPE bags) under solvent-free conditions. By optimizing the temperature and pressure of the photothermal reaction over a Ru/TiO$_2$ catalyst, LDPE and the other plastics could be converted into valuable liquid hydrocarbons with high selectivity. The photothermal system utilized UV light to activate polymeric chains, which were then cracked into lower molecular-weight molecules through the action of the supported Ru nanoparticles. Local heating of the Ru/TiO$_2$ catalyst under Vis and NIR light irradiation melted the polymers for intimate catalyst-polyolefin contact and provided reaction temperatures that enabled efficient C-C bond scission on Ru sites. At 1 bar $H_2/N_2$ (v/v = 70/30) and a reaction temperature of 300 °C, liquid/waxy fuels ($C_{27}$-centered distribution) and methane (selectivity >90% in gaseous products) were the main products. At 30 bar $H_2/N_2$ (v/v = 70/30) and 220 °C, valuable liquid fuels (86% gasoline- and diesel-range hydrocarbons, $C_5$-$C_{21}$) were obtained. Importantly, the photothermal catalytic recycling system worked efficiently under concentrated sunlight, paving the way for efficient solar-driven recycling of plastic wastes.

## Methods

### Synthesis of Ru/TiO$_2$ catalyst

Typically, 500 mg of TiO$_2$ powder (Degussa P25) and 4.11 mL of 5 mg/mL RuCl$_3$ (Aladdin) were added to a glass beaker containing deionized water (20 mL) under vigorous stirring. The resulting dispersion was heated at 60 °C for 2 h and then dried at 120 °C in an oil bath. The resulting Ru-impregnated TiO$_2$ powder was then reduced in a $H_2$/Ar (v/v = 10/90) atmosphere at 500 °C for 2 h in a tube furnace, using a heating rate of 5 °C/min. The obtained Ru/TiO$_2$ catalyst was then cooled to ambient temperature in a nitrogen atmosphere for subsequent catalytic tests.

### Contact angle measurements

A glass substrate coated with Ru/TiO$_2$ catalyst was prepared by spraying and drying. Then, a 2 × 2 × 1 mm (length-width-height) LDPE granule was placed on the as-prepared substrate and then heated to a designated temperature (120, 180, 240, or 300 °C) under an Ar atmosphere. After 5 min heating at the designated temperature, samples were then cooled to ambient temperature and static contact angle measurements were performed (OCA 20, Dataphysics, Germany). For the contact angles under Xe lamp or UV irradiation, identical steps were taken as the above operation except with light irradiation.

### Cross-sectional SEM

The cross-sectional SEM image and EDX maps were obtained on a S-4800 instrument (Hitachi, Japan) operating at a voltage of 10 kV. For the quenched molten LDPE-Ru/TiO$_2$ sample (state likes photothermal LDPE recycling), a mixture of Ru/TiO$_2$ and LDPE was heated at 300 °C under an Ar atmosphere for 30 min, then cooled to ambient temperature. For the physical mixture of the LDPE granule and Ru/TiO$_2$ sample (state likes traditional photocatalytic LDPE recycling), 10 mg of Ru/TiO$_2$ was first dispersed in anhydrous alcohol and sonicated for 20 min. Then, a 4 × 4 × 2 mm (length-width-height) LDPE granule was sprayed with Ru/TiO$_2$ ink and dried. The above two solidified samples

were then cut into pieces and sputtered with gold for cross-sectional SEM characterization studies.

## Photothermal catalytic polyolefin plastic recycling tests

For the polyolefin plastic recycling tests, polymer (80 mg) and $Ru/TiO_2$ (20 mg) were mixed to a uniform powder in an agate mortar. The obtained dark gray powder was then transferred into an ambient-pressure stainless reactor (with quartz lining and quartz windows). The reactor was sealed and then purged using alternating cycles of vacuum and Ar gas (six times). Next, the reactor was pressurized with a $H_2/Ar$ gas mixture (v/v = 30/70) to 1 bar. The dark gray powder was directly irradiated through the quartz window using a Xe lamp (Beijing Perfect-light Co. Ltd, PLS-SXE300) to achieve a specific reaction temperature and the temperature was maintained for a certain reaction time. The temperature could be controlled by varying the intensity of the Xe lamp. After the reaction, the reactor was allowed to cool naturally to ambient temperature. The gaseous products ($C_1$-$C_4$) formed were sampled using an airtight syringe and analyzed by gas chromatograph (Shimadzu, GC-2014C). The liquid/waxy products ($C_{5+}$) formed were extracted into cyclohexane with ultrasonication, after which the cyclohexane was removed by vacuum-rotary evaporation. The collected liquid/waxy products were analyzed by HTGC and $^1H$ NMR. Since the gaseous $C_1$-$C_4$ products could be precisely quantified, amounts of liquid/waxy products formed could be estimated as the mass of the reacted plastic minus the mass of the gaseous products. The actual yields of liquid/waxy products were also determined by accurate weighing (i.e., by weighing the quartz reactor lining and quartz window of the reactor before and after the reaction on a Sartorius balance, BSA224S-CW). The recovery rates were calculated based on the products collected versus the expected amount of products, being larger than 90% in all experiments reported herein. The insoluble residues (unreacted polymer and catalyst) after cyclohexane extraction of the liquid/waxy products were dried at 60 °C and weighed, then dissolved in TCB (containing BHT) for GPC analysis. Thermal polyolefin plastic recycling experiments were carried out in a manner similar to that described above for the photothermal recycling experiments, except that a heating element rather than a Xe lamp was used to achieve specific reaction temperatures. For the photolysis recycling system, procedures were similar to the photothermal recycling experiments. A Xe lamp equipped with a UV-Vis filter was used to irradiate the mixture of polymer and Ru-$TiO_2$ catalyst under 1 bar $H_2/Ar$ (v/v = 30/70) for 20 h, with the reaction temperature kept at 25 °C using circulating water cooling.

## High-pressure photothermal catalytic recycling experiments

The LDPE bags were frozen in liquid nitrogen and then pulverized. The pulverized LDPE bags (900 mg) and $Ru/TiO_2$ (100 mg) catalyst were mixed, then transferred into a high-pressure stainless reactor with a sapphire window. The reactor was sealed and then purged with alternating cycles of vacuum and nitrogen gas (six times). The reactor was then pressurized with $H_2/N_2$ (v/v = 70/30) to a designated pressure (10, 20, 30, and 40 bar). The dark gray LDPE-catalyst powder mixture was then irradiated with a Xe lamp (light intensity at 3.00 W cm$^{-2}$) and kept at a certain reaction temperature using auxiliary heating (as required) supplied by a heating element under continuous 800 rpm magnetic stirring. After the reaction, the reactor was cooled naturally to room temperature. The gaseous products were transferred to a vacuum vessel, then sampled with an airtight syringe and analyzed by gas chromatograph (Shimadzu, GC-2014C). The liquid/waxy products were extracted into $CH_2Cl_2$ with mesitylene as the internal standard. The gray slurry (containing liquid/waxy products and catalyst) was filtered by suction filtration and then analyzed by gas chromatograph (Shimadzu, GC-2014). The solid residues (low-solubility waxes, undegraded polymer residues, and catalyst) on the filter membrane were again extracted by hot cyclohexane (60 °C). The soluble compounds

were regarded as waxes, and insoluble compounds were catalyst and undegraded polymer residues. High-pressure thermal recycling experiments were carried using almost identical procedures as the high-pressure photothermal recycling but without Xe lamp irradiation.

## Data availability

The data supporting the findings of this study are available from the corresponding authors upon reasonable request. Source data are provided with this paper.

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

## Acknowledgements

The authors are grateful for financial support from the National Key Projects for Fundamental Research and Development of China (2018YFB1502002), the National Natural Science Foundation of China (51825205, 52120105002, 22102202, 22088102, and U22A20391), the DNL Cooperation Fund, CAS (DNL202016), the CAS Project for Young Scientists in Basic Research (YSBR-004), the Young Elite Scientist Sponsorship Program by CAST (2021QNRC001), and the Youth Innovation Promotion Association of the CAS. GINW acknowledges funding support from the MacDiarmid Institute for Advanced Materials and Nanotechnology and the Energy Education Trust of New Zealand. All NMR experiments were carried out at BioNMR facility, Tsinghua University Branch of China National Center for Protein Sciences (Beijing). We thank Dr. Ning Xu for assistance in NMR data collection.

## Author contributions

Y.M., Y.Z., and T.Z. conceived the idea for the project. Y.Z. and T.Z. supervised the project. Y.M. and Y.Z. conducted the measurement and characterizations. Y.M., Y.Z., G.W., R.S., L.W., and T.Z. wrote the manuscript. All authors discussed the results and commented on the manuscript.

## Competing interests

The authors declare no competing interests.
