## [Peer Review File · Nature Communications]

REVIEWER COMMENTS

Reviewer #1 (Remarks to the Author):

1. Author are encouraged to analyze the driving force behind change of contact angles at different reaction temperature. Would photochemical effects also affect in this process? Are there any differences between contact angles of photothermal melting/thermal melting?
2. Moreover, I notice remarkable promotion of degradation in Fig.S11 espically at low temperature. Could this be attributed to the plasmon-mediated catalysis of Ru nanopartilces? If so, the role of light is not merely limited in the activiation of LDPE chains and author is supposed to complete the mechanism in Fig.4f with further explanation of plasmon-mediated hydrogenolysis.
3. Author conducted contrast tests with Ru/TiO₂ and Ru powder catalysts to verify Ru nanoparticles are the main active sites. Catalytic hydrogenolysis of ploymer usually exhibites certain structure-sensitivity, while Fig.S2a and Fig.S17a gives obviously different size of Ru nanoparticles. Will that cause difference in their H₂ activation ability and consequently influence the distribution of products? To play safe, H₂ activation characterizations should be added to rule out its impact.
4. Have you considered the solar-to-fuel efficiency? It takes so long time for the reaction. How can you make it is feasibility in commercial application?
5. The temperature for the reaction is very difficult to measure. How can you make sure it is in the set temperature.
6. Finally, light intensity more than reaction temperature should be given in experimental details (including high-temperature photothermal experiments, which might demond higher concentrtrion).

Reviewer #2 (Remarks to the Author):

The authors report an interesting "photothermal" approach combined with catalytic "recycling" (more on that later) for transforming polyolefin plastics into liquid/waxy products under concentrated sunlight (xenon lamp) irradiation. This photothermal heating is carried out in the presence of a Ru/TiO₂ catalyst to 200-300 °C in the presence of the melted polyolefin. The authors also state this is "controlled hydrogenolysis" of C-C and C-H bonds in the polymer chains (mediated by Ru sites).

The paper contains interesting results that the rapidly growing circular economy-plastic upcycling community will find useful and interesting. Their "simple and novel strategy" however needs to be placed in the broader context of what has been demonstrated, to do so will require major revision. Here are my major points:

1. see lines 164/5 and the following three lines: These papers were not on "traditional thermal degradation", rather these papers are some of the very early hydrogenolysis papers (see conditions) on polyolefin upcycling to value added products. The authors are "cherry picking data" from these papers describing ONLY the control experiments reported (in the absence of hydrogen) !!
2. The authors choice of 1 bar H₂/Ar pressure is unfortunate because it is TOO low leading to the undesired LDPE, hydrogen poor residue they report.
3. The JACS paper by Enrique Iglesia (UCBerkeley) describes the probable and complex mechanism of Hydrogenolysis ...a series of dehydrogenation steps followed by hydrogenation leading to the dependence on the partial pressure/chemical potential of hydrogen to avoid what the author observe in (2).
4. There are many other contextual comments in the discussion that should be revisited in light of the recent literature on polyolefin upcycling to various value added liquids and fuels compared to the older pyrolysis and hydrocracking literature.
5. There is a lot of chemical engineering required if anyone will ever use photothermal energy at the scale needed for the large amount and variety of plastics dumped into the environment.

Reviewer #3 (Remarks to the Author):

This paper describes the application of photothermal hydrocracking of polyolefins which is a topic of considerable importance. The results are impressive (although the catalyst is not new), and the research is thorough with many control experiments used to confirm the key findings. The difference in activity between photothermal and thermal conditions is particularly clear from the data shown in fig 3c. Prior to publication some parts of the work need to be clarified.

Major points:

The thermal activity of the Ru/TiO₂ catalyst in the report (supplementary Fig 11.) is lower than reported in a previous study (<https://doi.org/10.1021/acscatal.1c00874>). The reaction conditions and catalysts loading used in the previous study should be evaluated to ensure the large benefit of the former and the reason for the low thermal activity of the catalyst better understood. The most impressive aspect of this paper is the difference between photothermal and thermal conditions and if one can achieve a similar activity without light by varying the pressure/temperature then the results are of less impact.

The authors state, 'UV light activates the LDPE chains, creating reaction sites for scission by Ru nanoparticles'. I do understand how activating the chains creates reaction sites. This needs to be clarified and some evidence presented. Also, it is unclear what activates the LDPE chains means, do they mean break to long chains into shorter chains? Some experimental evidence is required, e.g. the effect of UV without the catalyst, i.e.

UV light ☒ degrades polymer into oligomers

Ru/TiO₂ ☒ at 300°C degrade oligomers into liquids/gas alkanes

This could be verified by using model compounds (i.e. C16) for thermal degradation at 300°C?

The product selectivity should be investigated after repeated cycles of degradation (catalyst recycling) to determine if there are any changes in the product distribution.

It would be helpful to investigate the scalability of the reaction by exploring the feasibility of conducting the reaction on a larger scale, e.g. 5 - 10 g scale.

A recent paper in Nature Commun. described benchmarking of polyolefin catalysts and it would be interesting for the authors to benchmark their system using the method described that might also help better establish the mechanism of their system and roles of the Ru nanoparticles and TiO₂ support.

The systems works well under concentrated sunlight, which is excellent, but a control with 1 sun would be useful.

Minor points:

The title of the paper should reflect that it is not only light and heat that is required for the process, also gases are produced so the emphasis on liquids seems misplaced.

Figure 1. could be deleted, much of it is beyond the scope of the paper and key aspects, e.g. H₂, are missing.

Expressions like 'chopped up' are not scientific and should be replaced with serious scientific language. In general, more scientifically rigorous language would be welcome.

Reply to Reviewers' Comments

Comments from Reviewer #1:

Comment 1. Author are encouraged to analyze the driving force behind change of contact angles at different reaction temperature. Would photochemical effects also affect in this process? Are there any differences between contact angles of photothermal melting/thermal melting?

Response: We thank the reviewer for this kind suggestion. Since surface tension is temperature dependent for most thermoplastic polymers (e.g., polyolefins used in this manuscript), surface energy decreases with increasing temperature, resulting in a reduction of the static contact angle (θ)^[R1]. Hence, the contact angles between a molten LDPE droplet and Ru/TiO₂ decreased with increasing temperatures in the range of 120-300 °C. Besides, as shown in Supplementary Fig. 3, we observed a minor decrease of static contact angle of LDPE droplets under Xe lamp (i.e., photothermal condition, $\theta = 56^\circ$) or UV irradiation ($\theta = 57^\circ$) at 300 °C compared to data that without light irradiation (i.e., thermal condition, $\theta = 60^\circ$). The slightly decreased contact angle is likely due to the elevated mobility or reorientation of side chains or pendant groups on the macromolecules under UV irradiation^[R2,R3].

Supplementary Fig. 3 | Contact angles of LDPE droplet on Ru/TiO₂ with light irradiation. LDPE droplet contact angles on a glass substrate coated with Ru/TiO₂ under **a**, Xe lamp irradiation and **b**, UV irradiation at 300 °C in an argon atmosphere.

Comment 2. Moreover, I notice remarkable promotion of degradation in Fig.S11 especially at low temperature. Could this be attributed to the plasmon-mediated catalysis of Ru nanoparticles? If so, the role of light is not merely limited in the activation of LDPE chains and author is supposed to complete the mechanism in Fig.4f with further explanation of plasmon-mediated hydrogenolysis.

Response: Thank the reviewer for this valuable suggestion. Theoretical work by Moreno *et al.* showed that Ru metal has a bulk plasma frequency higher than 3 eV, and its local surface plasmon resonance (LSPR) may occur in the UV region^[R4]. Accordingly, we carried out 3-dimensional finite-difference time-domain (3D-FDTD)

simulations to determine the near-field intensities around the Ru nanoparticle loaded on the TiO₂ substrate. The simulations show that Ru nanoparticles had a maximum electric field intensity around them at 368 nm, close to the 310 nm in Moreno's calculated work (the difference mainly stems from different calculated parameters and nanoparticle size) (Supplementary Fig. 15a)^[R4]. Although 368 nm is very close to the exciting light 365 nm, the molecular weight distributions of polymeric residues under UV irradiation (Supplementary Fig. 16) were similar to that obtained under thermal conditions, excluding the possibility of enhanced degradation derived from the LSPR of the Ru nanoparticles. The absorption intensities of the Ru nanoparticle LSPR diminished as the Ru nanoparticle size decreased, almost disappearing at a diameter of 2 nm (cf. Ru size ~2.5 nm in our Ru/TiO₂ catalyst) (Supplementary Fig. 15b-d). Hence, both the experimental and theoretical evidence indicated that the remarkable promotion of degradation is not attributed to the LSPR of Ru nanoparticles.

Supplementary Fig. 15 | FDTD simulation of the Ru/TiO₂ catalyst. **a**, Normalized LSPR absorption intensity by FDTD simulation for Ru nanoparticles with different diameters (2 nm, 10 nm, or 50 nm) loaded on a TiO₂ substrate. **b-d**, Simulated electric fields around the Ru nanoparticles supported on TiO₂. The colored bar shows the electric field intensity ($\langle |E|^2 \rangle$).

Supplementary Fig. 16 | Exclusion of photocatalytic effect or LSPR effect over Ru/TiO₂. GPC analysis of molecular weight distributions of polymeric residues after thermal degradation (no 365 nm irradiation) or thermal degradation + 365 nm irradiation. Reaction times were 5 h in both experiments. For thermal degradation + 365 nm, $M_w = 56.0$ kDa, $\mathcal{D} = 10.4$; for thermal degradation, $M_w = 57.3$ kDa, $\mathcal{D} = 8.6$. Reaction conditions: 300 °C, 1 bar H₂/Ar (v/v = 30/70), 80 mg LDPE, 20 mg Ru/TiO₂.

Comment 3. Author conducted contrast tests with Ru/TiO₂ and Ru powder catalysts to verify Ru nanoparticles are the main active sites. Catalytic hydrogenolysis of polymer usually exhibits certain structure-sensitivity, while Fig.S2a and Fig.S17a gives obviously different size of Ru nanoparticles. Will that cause difference in their H₂ activation ability and consequently influence the distribution of products? To play safe, H₂ activation characterizations should be added to rule out its impact.

Response: We thank the reviewer for the meaningful questions. Specific surface area and Ru nanoparticle size will affect the H₂ adsorption ability and activation of the catalysts, thus accounting for the performance difference between Ru/TiO₂ and Ru powder (Fig. R1 and Supplementary Fig. 20). We have supplemented the corresponding discussion in the revised manuscript. Moreover, the lower performance of TiO₂ compared with that of Ru powder implied that Ru nanoparticles are the main active sites in Ru/TiO₂, though the difference was not significant. Hence, to verify our conclusion, we carried out the low-temperature hydrogenolysis experiments, which are catalyst-sensitivity, using n-hexadecane as a model compound (Supplementary Fig.21). The conversion of n-hexadecane on Ru powder was around 25 times higher than on TiO₂, validating that the Ru nanoparticles are the main active sites in Ru/TiO₂ for C-C bonds scission.

Fig. R1. TPD-H₂ spectra of Ru/TiO₂ and Ru powder.

Supplementary Fig. 20 | Thermal degradation of LDPE using different catalysts.

a, Degradation percentage and **b**, GPC analysis of molecular weight distributions of LDPE after thermal degradation with/without catalyst (Ru/TiO₂, Ru powder, TiO₂). Reaction conditions: 300 °C provided by heat element in the dark, 1 bar H₂/Ar (v/v = 30/70), 80 mg LDPE (100 mg for no-catalyst condition), 20 mg catalyst, reaction time 20 h.

Supplementary Fig. 21 | Thermal conversion of n-hexadecane. Thermal conversion

of n-hexadecane utilizing high-pressure reactor with different catalysts (*i.e.*, Ru/TiO₂, Ru powder, TiO₂). Conversion: 32.4% for Ru/TiO₂, 7.6% for Ru powder, and 0.3% for TiO₂. This experiment adopted a similar experimental procedure with the high-pressure thermal recycling of LDPE bags. Reaction conditions: 1600 mg n-hexadecane, 100 mg catalysts, 30 bar H₂/N₂ (v/v = 70/30), 240 °C, 5 h reaction.

Comment 4. *Have you considered the solar-to-fuel efficiency? It takes so long time for the reaction. How can you make it is feasibility in commercial application?*

Response: Thank you very much for raising this important point. Solar-to-fuel efficiency is an appropriate assessment for solar energy-to-chemical energy based on Gibbs free energy. Ismail *et al.* reported the Gibbs free energy of polyethylene (around 214-225 kJ/mol, changed with the decomposition extent) during its thermal decomposition, whereas it differs from our reaction systems and the products are also distinguished^[R5]. Given the complexity of products and reactants in the recycling reaction, precise evaluation of the Gibbs free energy is likely to go well beyond the scope of this manuscript, and we have to confess that it is challenging for us to accurately calculate the solar-to-fuel efficiency. The conversion of LDPE and selectivity of products may be more suitable to evaluate our system, according with other similar reactions (*e.g.*, photothermal Fischer-Tropsch synthesis^[R6] and photothermal CO₂ hydrogenation^[R7,R8]). Apologetically, we would consider the solar-to-fuel efficiency in future simpler photothermal recycling reactions of LDPE.

In this manuscript, we explored the feasibility of full-spectrum sunlight utilization in solar-driven recycling routes for plastic wastes. Although a good recycling performance is obtained, it still has a long way to go before practical application. In order to explore its feasibility in a commercial application, we carried out up-scaled photothermal recycling at a 5 g scale. The experimental results are also exciting, with 87% selectivity to gasoline- and diesel-range hydrocarbons (C₅-C₂₁) being obtained, close to the result at 900 mg scale (Supplementary Fig. 45). Furthermore, we attempted to perform a simple technoeconomic analysis of industrial polyethylene hydrogenolysis using Aspen Plus simulation software. Here, the industrial hydrogenolysis process of polyethylene was simulated by treating 8640 tons of polyethylene per year. In a typical process, the waste polyethylene, catalyst, and H₂ are pumped into the reactor (B1) with 100% of solid conversion to produce the C₁ to C₂₁ hydrocarbons, followed by the separation (B2) of gases including H₂ and C₁ to C₄ (6) and the nongaseous mixture of catalyst and C₅ to C₂₁ hydrocarbons (7). Subsequently, the gas components pass through separator 2 (B3) to isolate the light hydrocarbons (C₁ to C₄) (8), with excess H₂ being recycled and fed into the reactor for the next batch of reaction (3). A solid-liquid filter (B4) was used in another product line to separate the liquid C₅ to C₂₁ hydrocarbons (9), and the residual solid catalyst was recovered and reused (4). In addition, the liquids

were transported to a rectifying tower (B5) to vaporize the gasoline product (C₅ to C₁₂) (10) and isolate the diesel product (C₁₃ to C₂₂) (11). The energy consumed by each equipment and the total process was calculated on the basis of polyethylene hydrogenolysis by Ru/TiO₂. Compared to the two separators and the rectifying tower, the reactor consumes most of the energy (347.9 kW/h), accounting for 90.0 % of the energy input required for the whole process (Supplementary Table 3). Such energy consumption will be significantly lowered if the reactor is powered by solar energy using concentrated solar power technology^[R9], resulting in substantial cost reductions. Concentrated solar power technology has been reported commercially feasible in supercritical water gasification integrated with Fischer-Tropsch synthesis^[R10], liquid hydrocarbon fuels from CO₂ and H₂O^[R11], solar hydrogen production^[R9], liquid fuel and hydrogen coproduction^[R12]. In addition, molten salt thermal storage systems based on a tower design can achieve 24 h operation in the summertime^[R9]. Hence, our simple analysis suggests the commercial feasibility of photothermal polyolefin recycling due to the energy savings and rapid technological development of concentrated solar power technology. The capital investment cost of building a photothermal polyolefin recycling was not considered here.

Supplementary Fig. 45 | Up-scaled high-pressure photothermal recycling of LDPE bags. The FID signals of the liquid/waxy products formed during the high-pressure photothermal recycling of the LDPE bags under 30 bar H₂/N₂ (v/v = 70/30) at 220 °C for 5 h (CH₂Cl₂ was solvent, mesitylene was the internal standard). Reaction conditions: 500 mg Ru/TiO₂, 5000 mg LDPE bags. Inset, digital photographs of the filtered produced liquid fuels.

Supplementary Fig. 47 | Flow diagram of industrial polyethylene hydrogenolysis based on the simulation through Aspen Plus software.

Supplementary Table 3 | Energy consumption distribution of each component.

Component*	Energy consumption (kW/h)	Proportion (%)
Reactor	347.9	90.0
Separator 1	-6.5	-1.6
Rectifying tower	45.0	11.6
Total	386.4	100

* No energy consumption in separator 2 and filter.

Comment 5. The temperature for the reaction is very difficult to measure. How can you make sure it is in the set temperature.

Response: This is a good question. It is well-known that precise temperature detection in photothermal catalysis is a big challenge^[R13]. Currently, there are two common approaches to detecting reaction temperature in photothermal reactions: contact-type thermocouples based on the Seebeck effect^[R8,R14,R15], and non-contact-type infrared thermography based on blackbody radiation law^[R13]. In addition, FT-IR experiments of temperature-programmed adonitol transformation^[R16], or variations in the emission quantum yields and their emissive properties using quantum dots (QD) as temperature indicators^[R17] are also used to detect reaction temperature in some works. Zhang *et al.* also utilized the equilibrium state of the photothermal CO₂ hydrogenation reaction to infer the local temperature, but this method only adapts to reversible reactions^[R18]. We comprehensively consider the real-time monitoring and accessibility, and a K-type thermocouple that directly contact with the mixture of LDPE and Ru/TiO₂ catalyst may be the most suitable for temperature monitoring. As shown in Supplementary Fig. 5,

the temperature of the LDPE and Ru/TiO₂ catalyst mixture showed a linear correlation with the light intensity of the Xe lamp, thus allowing accurate control of the photothermal reaction temperature by adjustment of the light intensity. Moreover, the negligible temperature difference between infrared thermography and K-type thermocouple (304 °C vs. 300 °C) also verified our accurate detection (Fig. R2). In addition, we could also adjust the reaction temperature by auxiliary heating with a heating element or cooling using circulating water.

Supplementary Fig. 5 | Temperature at different light intensities. The temperature of a mixture of LDPE and Ru/TiO₂ catalyst under Xe lamp irradiation varied depending on the light intensity. The temperature was monitored by a K-type thermocouple in direct contact with the mixture LDPE and Ru/TiO₂ catalyst. Temperatures at specific light intensities were: 200 °C at 1.85 W cm⁻², 250 °C at 2.40 W cm⁻², 280 °C at 2.77 W cm⁻², 300 °C at 3.00 W cm⁻², 350 °C at 3.59 W cm⁻², 380 °C at 3.91 W cm⁻², and 400 °C at 4.11 W cm⁻².

Fig. R2. a, Digital photographs of the normal-pressure reactor. b, Infrared thermography for the normal-pressure reactor at 300 °C caused by 3.0 W cm⁻² Xe lamp irradiation.

Comment 6. Finally, light intensity more than reaction temperature should be given in experimental details (including high-temperature photothermal experiments, which might demand higher concentration).

Response: Thank you very much for your kind reminder. As the response in comment 5, the temperature of the LDPE and Ru/TiO₂ catalyst mixture showed a linear correlation with the light intensity of the Xe lamp, thus allowing accurate control of the photothermal reaction temperature by adjustment of the light intensity (Supplementary Fig. 5). Besides, the light intensities at different reactions were also supplemented in experimental details.

References:

- [R1] C. A. Fuentes, Y. Zhang, H. Guo, W. Woigk, K. Masania, C. Dransfeld, J. De Coninck, C. Dupont-Gillain, D. Seveno, A. W. Van Vuure, *Colloids Surf., A* **2018**, 558, 280.
- [R2] G. K. Belmonte, G. Charles, M. C. Strumia, D. E. Weibel, *Appl. Surf. Sci.* **2016**, 382, 93.
- [R3] F. Truica-Marasescu, P. Jedrzejowski, M. R. Wertheimer, *Plasma Processes Polym.* **2004**, 1, 153.
- [R4] J. M. Sanz, D. Ortiz, R. Alcaraz de la Osa, J. M. Saiz, F. González, A. S. Brown, M. Losurdo, H. O. Everitt, F. Moreno, *J. Phys. Chem. C* **2013**, 117, 19606.
- [R5] N. Rasaidi, A. R. Mohamad Daud, S. N. Ismail, *Int. J. Renew. Energy Dev.* **2022**, 11, 829.
- [R6] L. Song, S. Ouyang, P. Li, J. Ye, *J. Mater. Chem. A* **2022**, 10, 16243.
- [R7] M. Sun, B. Zhao, F. Chen, C. Liu, S. Lu, Y. Yu, B. Zhang, *Chem. Eng. J.* **2021**, 408, 127280.
- [R8] L. Liu, A. V. Puga, J. Cored, P. Concepción, V. Pérez-Dieste, H. García, A. Corma, *Appl. Catal., B* **2018**, 235, 186.
- [R9] N. Monnerie, H. von Storch, A. Houaijia, M. Roeb, C. Sattler, *Int. J. Hydrogen Energy* **2017**, 42, 13498.
- [R10] A. Rahbari, A. Shirazi, M. B. Venkataraman, J. Pye, *Energy Convers. Manage.* **2019**, 184, 636.
- [R11] R. Schäppi, D. Rutz, F. Dähler, A. Muroyama, P. Haueter, J. Lilliestam, A. Patt, P. Furler, A. Steinfeld, *Nature* **2022**, 601, 63.
- [R12] F. He, J. Trainham, G. Parsons, J. S. Newman, F. Li, *Energy Environ. Sci.* **2014**, 7, 2033.
- [R13] L. Mascaretti, A. Schirato, T. Montini, A. Alabastri, A. Naldoni, P. Fornasiero, *Joule* **2022**, 6, 1727.
- [R14] Z. Li, J. Liu, Y. Zhao, G. I. N. Waterhouse, G. Chen, R. Shi, X. Zhang, X. Liu, Y. Wei, X.-D. Wen, L.-Z. Wu, C.-H. Tung, T. Zhang, *Adv. Mater.* **2018**, 30, 1800527.
- [R15] K. K. Ghuman, T. E. Wood, L. B. Hoch, C. A. Mims, G. A. Ozin, C. V. Singh, *Phys. Chem. Chem. Phys.* **2015**, 17, 14623.
- [R16] C. Mao, H. Li, H. Gu, J. Wang, Y. Zou, G. Qi, J. Xu, F. Deng, W. Shen, J. Li, S. Liu, J. Zhao, L. Zhang, *Chem* **2019**, 5, 2702.

- [R17] D. Mateo, J. Albero, H. García, *Energy Environ. Sci.* **2017**, *10*, 2392.
- [R18] M. Cai, Z. Wu, Z. Li, L. Wang, W. Sun, A. A. Tountas, C. Li, S. Wang, K. Feng, A.-B. Xu, S. Tang, A. Tavasoli, M. Peng, W. Liu, A. S. Helmy, L. He, G. A. Ozin, X. Zhang, *Nat. Energy* **2021**, *6*, 807.

Comments from Reviewer #2:

General Comments: *The authors report an interesting "photothermal" approach combined with catalytic "recycling" (more on that later) for transforming polyolefin plastics into liquid/waxy products under concentrated sunlight (xenon lamp) irradiation. This photothermal heating is carried out in the presence of a Ru/TiO₂ catalyst to 200-300 °C in the presence of the melted polyolefin. The authors also state this is "controlled hydrogenolysis" of C-C and C-H bonds in the polymer chains (mediated by Ru sites). The paper contains interesting results that the rapidly growing circular economy-plastic upcycling community will find useful and interesting. Their "simple and novel strategy" however needs to be placed in the broader context of what has been demonstrated, to do so will require major revision. Here are my major points:*

Response: We thank the reviewer for the positive comments. All the issues raised by the reviewer have been carefully considered and addressed in the revised manuscript.

Comment 1. *see lines 164/5 and the following three lines: These papers were not on "traditional thermal degradation", rather these papers are some of the very early hydrogenolysis papers (see conditions) on polyolefin upcycling to value added products. The authors are "cherry picking data" from these papers describing ONLY the control experiments reported (in the absence of hydrogen) !!*

Response: Thank the reviewer for this kind reminder. We sincerely apologize for our carelessness, and we have deleted the word "traditional". For photothermal degradation, the weight-average molecular weight (M_w) of the LDPE determined by gel permeation chromatography (GPC) dropped from 69.8 kDa (for pristine LDPE) to 12.9 kDa after 1 h at 300 °C (Supplementary Fig. 8). On extending the reaction time to 10 h, the M_w dropped further to 3.4 kDa, implying the efficient scission of C-C bonds in the LDPE polymeric chains. Besides, we supplemented the dispersity of polymer residues at different thermal reaction times by GPC analysis. As shown in Supplementary Fig. 9, thermal degradation of LDPE over Ru/TiO₂ exhibit large M_w and wide dispersity ($D = M_w/M_n$) in our work probably due to the random scission of C-C bonds and insufficient cracking of the high-molecular-weight portion of polymer residues. The corresponding data and discussions have now been addressed in the revised manuscript.

Supplementary Fig. 8 | GPC analysis of polymer residues at different photothermal reaction times. **a**, Molecular weight distribution and **b**, Dispersity (D) plots of the polymer residues at different photothermal reaction times. $M_{w-1h} = 12.9$ kDa, $M_{w-5h} = 4.7$ kDa, $M_{w-10h} = 3.4$ kDa. Corresponding dispersities were $D_{1h} = 5.9$, $D_{5h} = 3.4$, and $D_{10h} = 2.3$, respectively. Reaction conditions: 1 bar H_2/Ar (v/v = 30/70), 300 °C (provided by 3.0 $W\ cm^{-2}$ Xe lamp), 80 mg LDPE, 20 mg Ru/TiO₂.

Supplementary Fig. 9 | GPC analysis of polymer residues at different thermal reaction times. **a**, Molecular weight distribution and **b**, Dispersity (D) plots of the polymer residues at different thermal reaction times. $M_{w-5h} = 57.3$ kDa, $M_{w-10h} = 46.1$ kDa, $M_{w-20h} = 25.0$ kDa. Corresponding dispersities were $D_{5h} = 8.6$, $D_{10h} = 6.9$, and $D_{20h} = 4.9$, respectively. Reaction conditions: 1 bar H_2/Ar (v/v = 30/70), 300 °C (provided by heating element), 80 mg LDPE, 20 mg Ru/TiO₂.

Comment 2. *The authors choice of 1 bar H_2/Ar pressure is unfortunate because it is TOO low leading to the undesired LDPE, hydrogen poor residue they report.*

Response: Thanks for your valuable comment and we are sorry for the confusion. The low hydrogen partial pressure causes a deficiency of both the chemisorbed hydrogen atoms and the related intermediates that cleaves the C–C bonds, thus the low activity

in the LDPE hydrogenolysis reaction. Considering that low-pressure reactions are more accessible to be performed and to study the reaction mechanism, we first chose low pressure as the reaction condition. Upon finding the advantages of the photothermal system compared with the thermal system, we further expanded it to high-pressure reactions.

***Comment 3.** The JACS paper by Enrique Iglesia (UCBerkeley) describes the probable and complex mechanism of Hydrogenolysis ...a series of dehydrogenation steps followed by hydrogenation leading to the dependence on the partial pressure/chemical potential of hydrogen to avoid what the author observe in (2).*

Response: This is an insightful point. We also noticed the importance of sufficient hydrogen partial pressure. Hence, we carried out the pressure-dependent photothermal reactions to explore the influence of the hydrogen partial pressure. At a pressure of 10 bar, a substantial amount of waxy products was noticed due to poor hydrogenolysis under low hydrogen partial pressure (Fig. 4a). The selectivity of waxy products decreased and liquid fuels gradually increased as the pressure elevated. And the distribution of the liquid/waxy products shifted towards lower carbon numbers as the reaction pressure increased in the range of 10 bar to 30 bar, while it remained almost unchanged with the pressure increasing to 40 bar (Fig. 4c). Hence, low hydrogen partial pressures (e.g., below 10 bar) yield substantial amounts of wax products, whilst sufficiently high hydrogen partial pressures (e.g., 20 to 30 bar) result in high selectivity to liquid fuels.

Fig. 4 | Pressure-dependent product distributions of photothermal polyolefin plastic recycling system. a, Product selectivities of the high-pressure (220 °C for 3 h over Ru/TiO₂ catalyst) photothermal recycling of LDPE bags. **b**, Process of high-pressure photothermal recycling of LDPE bags under 30 bar H₂/N₂ (v/v = 70/30). **c**, Distributions of liquid products obtained from the photothermal recycling of LDPE bags at 220 °C for 3 h at different reaction pressures (10, 20, 30, 40 bar H₂/N₂ (v/v = 70/30)) over Ru/TiO₂.

Comment 4. There are many other contextual comments in the discussion that should be revisited in light of the recent literature on polyolefin upcycling to various value added liquids and fuels compared to the older pyrolysis and hydrocracking literature.

Response: Thank the reviewer for this very useful suggestion. We have revised contextual comments in the revised manuscript, and explained some experimental results by previous studies. For instance, several classic studies by Iglesia *et al.* and

Hibbitts *et al.* were cited to explain the influence of reaction pressure^[R19,R20]. Moreover, a recent benchmarking by Dyson *et al.* was cited to verify that Ru nanoparticles are the main active sites and hydrogenolysis is the main reaction route^[R21]. In addition, the reported explanation of methanation over Ru sites by Vlachos *et al.* was also cited to explain our high-selectivity methane in gaseous products under normal pressure^[R22], *etc.*

Comment 5. *There is a lot of chemical engineering required if anyone will ever use photothermal energy at the scale needed for the large amount and variety of plastics dumped into the environment.*

Response: We thank the reviewer for the constructive suggestion. In this manuscript, we preliminarily explored the feasibility of full-spectrum sunlight utilization in solar-driven recycling routes for plastic wastes. Although a good recycling performance is obtained, it still has a long way to go before practical application. In order to explore its feasibility in a commercial application, we carried out up-scaled photothermal recycling at a 5 g scale. The measured results are also exciting and 87% selectivity of gasoline- and diesel-range hydrocarbons (C₅-C₂₁) is obtained, close to the result at 900 mg scale (Supplementary Fig. 45). Furthermore, we attempted to perform a simple technoeconomic analysis of industrial polyethylene hydrogenolysis using Aspen Plus simulation software. Here, the industrial hydrogenolysis process of polyethylene was simulated by treating 8640 tons of polyethylene per year. In a typical process, the waste polyethylene, catalyst, and H₂ are pumped into the reactor (B1) with 100% of solid conversion to produce the C₁ to C₂₁ hydrocarbons, followed by the separation (B2) of gases including H₂ and C₁ to C₄ (6) and the nongaseous mixture of catalyst and C₅ to C₂₁ hydrocarbons (7). Subsequently, the gas components pass through separator 2 (B3) to isolate the light hydrocarbons (C₁ to C₄) (8), with excess H₂ being recycled and fed into the reactor for the next batch of reaction (3). A solid-liquid filter (B4) was used in another product line to separate the liquid C₅ to C₂₁ hydrocarbons (9), and the residual solid catalyst was recovered and reused (4). In addition, the liquids were transported to a rectifying tower (B5) to vaporize the gasoline product (C₅ to C₁₂) (10) and isolate the diesel product (C₁₃ to C₂₂) (11). The energy consumed by each equipment and the total process was calculated on the basis of polyethylene hydrogenolysis by Ru/TiO₂. Compared to the two separators and the rectifying tower, the reactor consumes most of the energy (347.9 kW/h), accounting for 90.0 % of the energy input required for the whole process (Supplementary Table 3). Such energy consumption will be significantly lowered if the reactor is powered by solar energy using concentrated solar power technology^[R9], resulting in substantial cost reductions. Concentrated solar power technology has been reported commercially feasible in supercritical water gasification integrated with Fischer-Tropsch synthesis^[R10], liquid hydrocarbon fuels from CO₂ and

H_2O ^[R11], solar hydrogen production^[R9], liquid fuel and hydrogen coproduction^[R12]. In addition, molten salt thermal storage systems based on a tower design can achieve 24 h operation in the summertime^[R9]. Hence, our simple analysis suggests the commercial feasibility of photothermal polyolefin recycling due to the energy savings and rapid technological development of concentrated solar power technology. The capital investment cost of building a photothermal polyolefin recycling was not considered here.

Supplementary Fig. 45 | Up-scaled high-pressure photothermal recycling of LDPE bags. The FID signals of the liquid/waxy products formed during the high-pressure photothermal recycling of the LDPE bags under 30 bar H_2/N_2 ($v/v = 70/30$) at 220 °C for 5 h (CH_2Cl_2 was solvent, mesitylene was the internal standard). Reaction conditions: 500 mg Ru/ TiO_2 , 5000 mg LDPE bags. Inset, digital photographs of the filtered produced liquid fuels.

Supplementary Fig. 47 | Flow diagram of industrial polyethylene hydrogenolysis based on the simulation through Aspen Plus software.

Supplementary Table 3 | Energy consumption distribution of each component.

Component*	Energy consumption (kW/h)	Proportion (%)
Reactor	347.9	90.0
Separator 1	-6.5	-1.6
Rectifying tower	45.0	11.6
Total	386.4	100

* No energy consumption in separator 2 and filter.

References:

- [R9] N. Monnerie, H. von Storch, A. Houaijia, M. Roeb, C. Sattler, *Int. J. Hydrogen Energy* **2017**, *42*, 13498.
- [R10] A. Rahbari, A. Shirazi, M. B. Venkataraman, J. Pye, *Energy Convers. Manage.* **2019**, *184*, 636.
- [R11] R. Schächli, D. Rutz, F. Dähler, A. Muroyama, P. Haueter, J. Lilliestam, A. Patt, P. Furler, A. Steinfeld, *Nature* **2022**, *601*, 63.
- [R12] F. He, J. Trainham, G. Parsons, J. S. Newman, F. Li, *Energy Environ. Sci.* **2014**, *7*, 2033.
- [R19] D. W. Flaherty, E. Iglesia, *J. Am. Chem. Soc.* **2013**, *135*, 18586.
- [R20] A. Almithn, D. Hibbitts, *J. Phys. Chem. C* **2019**, *123*, 5421.
- [R21] W.-T. Lee, A. van Muyden, F. D. Bobbink, M. D. Mensi, J. R. Carullo, P. J. Dyson, *Nat. Commun.* **2022**, *13*, 4850.
- [R22] C. Wang, K. Yu, B. Sheludko, T. Xie, P. A. Kots, B. C. Vance, P. Kumar, E. A. Stach, W. Zheng, D. G. Vlachos, *Appl. Catal., B* **2022**, *319*, 121899.

Comments from Reviewer #3:

General Comments: *This paper describes the application of photothermal hydrocracking of polyolefins which is a topic of considerable importance. The results are impressive (although the catalyst is not new), and the research is thorough with many control experiments used to confirm the key findings. The difference in activity between photothermal and thermal conditions is particularly clear from the data shown in fig 3c. Prior to publication some parts of the work need to be clarified.*

Response: Thanks for the positive comments. We carefully considered those suggestions and modified our manuscript accordingly.

Major points:

Comment 1. *The thermal activity of the Ru/TiO₂ catalyst in the report (supplementary Fig 11.) is lower than reported in a previous study (<https://doi.org/10.1021/acscatal.1c00874>). The reaction conditions and catalysts loading used in the previous study should be evaluated to ensure the large benefit of the former and the reason for the low thermal activity of the catalyst better understood. The most impressive aspect of this paper is the difference between photothermal and thermal conditions and if one can achieve a similar activity without light by varying the pressure/temperature then the results are of less impact.*

Response: We thank the reviewer for the thoughtful comment. The thermal activity of the Ru/TiO₂ catalyst in Supplementary Fig. 14 is lower than that in the study by Vlachos *et al*, which would be due to the huge difference of hydrogen partial pressure that normal pressure was used in the former experiment while high pressure (30 bar H₂) was utilized in Vlachos's study. Under high-pressure photothermal recycling, the thermal activity of the Ru/TiO₂ is close to the measured results in Vlachos's work. To ensure the large benefit of our systems, we synthesized Ru/TiO₂ (5.0 wt.% Ru) that is similar to the previous study^[R23], using the same impregnation method as our Ru/TiO₂ catalyst (2.0 wt.% Ru). Identical reaction conditions were used to compare the thermal and photothermal recycling of isotactic PP ($M_w = 1.2$ kDa) expect that the 40 bar H₂/N₂ (v/v = 70/30) mixed gas since high-purity hydrogen control of our institute (30 bar H₂ was used in Vlachos's work). After thermal recycling, similar products distribution (21% gas, 55% liquid, 24% solid) compared with the reported results (10% gas, 70% liquid, 20% solid) was noticed (the difference may be due to distinction of the reactor and reaction atmosphere), verifying the close thermal activity of our Ru/TiO₂ catalyst with previous work (Supplementary Fig. 44). After the photothermal recycling, no solid was witnessed and achieving close gas selectivity but higher liquid selectivity (25% gas, 75% liquid) compared with that of thermal recycling, further indicating the advantage of photothermal recycling.

To further study the difference between photothermal and thermal recycling, we carried out thermal recycling of LDPE bags at different reaction times (Supplementary Fig. 38 and Supplementary Table 2). Results showed that a longer reaction time (thermal: 9 h vs. photothermal: 3 h) was required in thermal recycling to realize similar selectivity of liquid fuels with that of photothermal recycling, while 7% wax was still present. Although the wax was completely converted as further prolonging reaction time to 12 h, the selectivity of liquid fuels steeply declined and gas increased. Hence, based on the perspective of energy consumption and product selectivity, we can verify the huge advantage and significant meaning of photothermal polyolefin recycling.

Supplementary Fig. 44 | High-pressure photothermal recycling of isotactic PP. Product selectivities of the high-pressure recycling of isotactic PP ($M_w = 1.2$ kDa, Sigma-Aldrich) under photothermal or thermal conditions. For thermal recycling: 21% gas, 55% liquid, 24% solid; for photothermal recycling: 25% gas, 75% liquid. Reaction conditions: 5.0 wt.% Ru/TiO₂ (50 mg), isotactic PP (2000 mg), 250 °C, reaction time 6 h, 40 bar H₂/N₂ (v/v = 70/30).

Supplementary Fig. 38 | Thermal recycling of LDPE bags at different reaction times. Product distributions of thermal recycling of LDPE bags at different reaction

times. Reaction conditions: 220 °C without Xe lamp irradiation, 30 bar H₂/N₂ (v/v = 70/30), 900 mg pulverized LDPE bags, 100 mg Ru/TiO₂ catalyst.

Supplementary Table 2 | Product distributions of high-pressure thermal recycling of LDPE bags. TD represents thermal degradation recycling. Reaction conditions: 220 °C, 30 bar H₂/N₂ (v/v = 70/30), 900 mg pulverized LDPE bags, 100 mg Ru/TiO₂ catalyst.

LDPE bag → Gas (C ₁ -C ₄) + liquid fuel (C ₅ -C ₂₁) + wax (C ₂₂₊)						
Entry	Type	Time (h)	R _d (%)	Selectivity (%)		
				Gas (C ₁ -C ₄)	Liquid fuels (C ₅ -C ₂₁)	Wax (C ₂₂₊)*
1	TD	3	97	10	64	26
2	TD	6	100	11	70	19
3	TD	9	100	13	80	7
4	TD	12	100	26	74	0

* The amount was determined by the C₂₂₊ products measured by gas chromatograph and the mass of low-solubility wax. R_d represents the degradation percentage.

Comment 2. The authors state, ‘UV light activates the LDPE chains, creating reaction sites for scission by Ru nanoparticles’. I do understand how activating the chains creates reaction sites. This needs to be clarified and some evidence presented. Also, it is unclear what activates the LDPE chains means, do they mean break to long chains into shorter chains? Some experimental evidence is required, e.g. the effect of UV without the catalyst, i.e. UV light degrades polymer into oligomers, Ru/TiO₂ at 300°C degrade oligomers into liquids/gas alkanes. This could be verified by using model compounds (i.e. C16) for thermal degradation at 300°C?

Response: Thanks for your professional comment. According to the review’s suggestions, we supplemented corresponding experiments. As shown in Supplementary Fig. 17 and Fig. R3, we found that UV light is indeed able to promote the polymer converting into oligomers ($M_w = 670$ Da, $\bar{D} = 2.2$), since the degradation can be promoted from 1.3% of thermal condition to 10.0% of photothermal condition. The Ru/TiO₂ also displayed well for the hydrogenolysis of n-hexadecane (32.4% conversion to lower carbon number) (Fig. R4). The tandem routes, in which UV promotes the conversion of polymer into oligomers, and then the Ru/TiO₂ further converts the oligomers into lower carbon-number hydrocarbons, seem to be tenable. But since Ru/TiO₂ can also convert polymer into oligomers (7.8% conversion), the rate-determining step is more likely to be the activation of polymer chains. Besides, from the perspective of the amount of reactants, only 10.0% of the LDPE occurred conversion to oligomers, but 90.0% of the undegraded polymer residues remained ($M_w = 11.4$ kDa and $\bar{D} = 4.3$) (Supplementary Fig. 17). Hence, a route that UV directly acts on inert

polymer instead of converting polymer into oligomers may be more acceptable and efficient. Furthermore, we detected the selectivity of gaseous products in LDPE decomposition without any catalyst with/without UV irradiation (Supplementary Fig. 18). Intriguingly, we found that CH₄ production was greatly suppressed after the introduction of UV light. Previous studies of LDPE thermal decomposition have shown that CH₄ generation results from direct terminal C-C scission (methane produced *via* a surface cascade of consecutive C-C scissions can be nearly ignored without any catalyst at this reaction temperature)^[R22]. Lower production rates of CH₄ under UV irradiation indicated preferential internal C-C scission instead of terminal C-C scission, implying that internal C-C bonds were more easily weakened by UV irradiation, and thus changing the cleavage mode^[R24]. Hence, quicker degradation of LDPE and higher selectivity of liquid fuels can be achieved using the photothermal recycling. We are deeply aware of the importance of the reaction mechanism, and we are actively looking for collaborators with expertise in this area to assist our future work.

Supplementary Fig. 17 | Direct irradiation of LDPE without any catalyst. a, Degradation percentage and **b,** GPC molecular weight distributions of pristine LDPE or LDPE after photothermal or thermal degradation experiments without any catalyst. Reaction conditions: 300 °C provided by Xe lamp and auxiliary heating (photothermal), or only heating in the dark (thermal), 1 bar H₂/Ar (v/v = 30/70), 100 mg LDPE, reaction time 20 h.

Fig. R3. GPC analysis of liquid/waxy products of photothermal degradation of LDPE without any catalyst. $M_w = 670$ Da, $D = 2.2$. Reaction conditions: 300 °C provided by Xe lamp and auxiliary heating (photothermal), 1 bar H₂/Ar (v/v = 30/70), 100 mg LDPE, reaction time 20 h.

Fig. R4. The FID signals of the gas chromatography in the thermal conversion of n-hexadecane over Ru/TiO₂. Reaction conditions: 1600 mg n-hexadecane, 100 mg Ru/TiO₂, 30 bar H₂/N₂ (v/v = 70/30), 240 °C, 5 h reaction.

Supplementary Fig. 18 | Effects of UV irradiation on LDPE. **a**, FID signals of the gas chromatography. **b**, Selectivity to gaseous products after thermal degradation of LDPE in a quartz lining without any catalyst under UV irradiation (Thermal + UV) or

no UV irradiation (Thermal). Reaction conditions: 220 °C, 20 bar H₂/N₂ (v/v = 70/30), 700 mg LDPE, reaction time 3 h. c, Proposed LDPE cleavage modes with or without UV irradiation.

Comment 3. *The product selectivity should be investigated after repeated cycles of degradation (catalyst recycling) to determine if there are any changes in the product distribution.*

Response: Thank the reviewer for this kind reminder. We have given the reusability measurements of the Ru/TiO₂ catalyst for photothermal LDPE degradation in Supplementary Fig. 11. Close selectivity of gaseous products, and similar distribution of liquid/waxy products were noticed of the photothermal catalytic degradation of LDPE over the reused Ru/TiO₂ catalyst, suggesting the good repeatability of the photothermal catalytic recycling system (Supplementary Fig. 28 and 35).

Supplementary Fig. 11 | Reusability of the Ru/TiO₂ catalyst for photothermal LDPE degradation. Reaction conditions: 300 °C, 1 bar H₂/Ar (v/v = 30/70), 80 mg LDPE, 20 mg Ru/TiO₂ (reused), reaction time 20 h for each cycle.

Supplementary Fig. 28 | Gaseous products of photothermal catalytic degradation of LDPE over the reused Ru/TiO₂ catalyst. a, Selectivity and b, product yields of photothermal catalytic degradation of LDPE. Reaction conditions: 300 °C, 1 bar H₂/Ar (v/v = 30/70), 80 mg LDPE, 20 mg reused Ru/TiO₂, reaction time of 20 h.

Supplementary Fig. 35 | Liquid/waxy products of photothermal catalytic

degradation of LDPE over the reused Ru/TiO₂ catalyst. a-d, HTGC of the isolated liquid/waxy products for the first, second, third, and fourth measurements over the reused Ru/TiO₂ catalyst, respectively. Reaction conditions: 300 °C, 1 bar H₂/Ar (v/v = 30/70), 80 mg LDPE, 20 mg reused Ru/TiO₂, reaction time of 20 h.

Comment 4. *It would be helpful to investigate the scalability of the reaction by exploring the feasibility of conducting the reaction on a larger scale, e.g. 5 - 10 g scale.*

Response: We thank the reviewer for the valuable suggestion. In order to explore its feasibility in a scaled application, we carried out up-scaled photothermal recycling at a 5 g scale. The results were also exciting and 87% selectivity of gasoline- and diesel-range hydrocarbons (C₅-C₂₁) was obtained, which is close to the result at 900 mg scale (Supplementary Fig. 45).

Supplementary Fig. 45 | Up-scaled high-pressure photothermal recycling of LDPE bags. The FID signals of the liquid/waxy products formed during the high-pressure photothermal recycling of the LDPE bags under 30 bar H₂/N₂ (v/v = 70/30) at 220 °C for 5 h (CH₂Cl₂ was the solvent, mesitylene was the internal standard). Reaction conditions: 500 mg Ru/TiO₂, 5000 mg LDPE bags. Inset, digital photograph of the filtered liquid products.

Comment 5. *A recent paper in Nature Commun. described benchmarking of polyolefin catalysts and it would be interesting for the authors to benchmark their system using the method described that might also help better establish the mechanism of their system and roles of the Ru nanoparticles and TiO₂ support.*

Response: Thank you very much for your valuable comment. The reaction temperatures in Dyson's work are 275, 325, 375 °C, which would be too high for our photothermal reactor (a maximum temperature of 250 °C). However, according to the key idea of the recent benchmarking, we carried out the hydrogenolysis experiments using n-hexadecane as a model compound (Supplementary Fig. 21)^[R21]. The experimental results demonstrate that around 25 times higher conversion of n-

hexadecane on Ru powder can be noticed compared with that on TiO₂, validating the Ru nanoparticles as the main active sites and hydrogenolysis as the main reaction route.

Supplementary Fig. 21 | Thermal conversion of n-hexadecane. Thermal conversion of n-hexadecane utilizing a high-pressure reactor with different catalysts (*i.e.*, Ru/TiO₂, Ru powder, TiO₂). Conversion: 32.4% for Ru/TiO₂, 7.6% for Ru powder, and 0.3% for TiO₂. This experiment adopted a similar experimental procedure with the high-pressure thermal recycling of LDPE bags. Reaction conditions: 1600 mg n-hexadecane, 100 mg catalysts, 30 bar H₂/N₂ (v/v = 70/30), 240 °C, 5 h reaction.

Comment 6. *The systems works well under concentrated sunlight, which is excellent, but a control with 1 sun would be useful.*

Response: Thank you very much for raising this valuable suggestion. Given that the remarkable promotion of degradation was mainly due to synergistic utilization of UV, Vis, and NIR light, the light intensity of the irradiation can determine the degradation efficiency. We utilized identical reaction temperatures by auxiliary heating but different light intensities to study the photothermal degradation of the LDPE over Ru/TiO₂. It can be found that the degradation rate decreases as the light intensity reduces, further implying the importance of sufficient light irradiation (R_d is 10.5% under the light intensity of 0.1 W cm⁻²) (Fig. R5). Since the low light intensity of unconcentrated sunlight, researchers often concentrate it to realize higher light intensity or local heating^[R25,R26]. Besides, the concentrated solar power technology has been reported commercially feasible in supercritical water gasification integrated with Fischer-Tropsch synthesis^[R10], liquid hydrocarbon fuels from CO₂ and H₂O^[R11], solar hydrogen production^[R9], liquid fuel and hydrogen coproduction^[R12], *etc.* In addition, molten salt thermal storage systems based on a tower design can achieve 24 h operation in the summertime^[R9].

Fig. R5. Photothermal degradation of LDPE over Ru/TiO₂ catalyst under different light intensities. Reaction conditions: 300 °C, 1 bar H₂/Ar (v/v = 30/70), 80 mg LDPE, 20 mg Ru/TiO₂, reaction time 20 h.

Minor points:

Comment 7. *The title of the paper should reflect that it is not only light and heat that is required for the process, also gases are produced so the emphasis on liquids seems misplaced.*

Response: Thanks for the insightful suggestion. We have revised the title (Photothermal recycling of waste polyolefin plastics into liquid fuels with high selectivity under mild conditions) in the revised manuscript.

Comment 8. *Figure 1. could be deleted, much of it is beyond the scope of the paper and key aspects, e.g. H₂, are missing.*

Response: Thank you very much for your kind reminder. We have deleted this figure in the revised manuscript.

Comment 9. *Expressions like ‘chopped up’ are not scientific and should be replaced with serious scientific language. In general, more scientifically rigorous language would be welcome.*

Response: Thank you very much for this suggestion. We have revised this expression in the revised manuscript.

References:

- [R9] N. Monnerie, H. von Storch, A. Houaijia, M. Roeb, C. Sattler, *Int. J. Hydrogen Energy* **2017**, *42*, 13498.
- [R10] A. Rahbari, A. Shirazi, M. B. Venkataraman, J. Pye, *Energy Convers. Manage.* **2019**, *184*, 636.
- [R11] R. Schäppi, D. Rutz, F. Dähler, A. Muroyama, P. Haueter, J. Lilliestam, A. Patt,

- P. Furler, A. Steinfeld, *Nature* **2022**, *601*, 63.
- [R12] F. He, J. Trainham, G. Parsons, J. S. Newman, F. Li, *Energy Environ. Sci.* **2014**, *7*, 2033.
- [R21] W.-T. Lee, A. van Muyden, F. D. Bobbink, M. D. Mensi, J. R. Carullo, P. J. Dyson, *Nat. Commun.* **2022**, *13*, 4850.
- [R22] C. Wang, K. Yu, B. Sheludko, T. Xie, P. A. Kots, B. C. Vance, P. Kumar, E. A. Stach, W. Zheng, D. G. Vlachos, *Appl. Catal., B* **2022**, *319*, 121899.
- [R23] P. A. Kots, S. Liu, B. C. Vance, C. Wang, J. D. Sheehan, D. G. Vlachos, *ACS Catal.* **2021**, *11*, 8104.
- [R24] B. C. Vance, P. A. Kots, C. Wang, J. E. Granite, D. G. Vlachos, *Appl. Catal., B* **2023**, *322*, 122138.
- [R25] Y. Liu, Q. Zhong, P. Xu, H. Huang, F. Yang, M. Cao, L. He, Q. Zhang, J. Chen, *Matter* **2022**, *5*, 1305.
- [R26] Z. Li, X. Zhang, J. Liu, R. Shi, G. I. N. Waterhouse, X. Wen, T. Zhang, *Adv. Mater.* **2021**, 2103248.

REVIEWERS' COMMENTS

Reviewer #1 (Remarks to the Author):

The revision is well done but the claim that the Photothermal recycling is under mild condition should be reconsidered. The pressure is from 10bar to 40bar, which is not so mild.

Reviewer #2 (Remarks to the Author):

The extensive response to comments of the three reviewers and revisions are fine and the paper can be published.

Reviewer #3 (Remarks to the Author):

The authors have extensively revised the manuscript and thoroughly addressed the key issues raised by the reviewers. I am pleased to accept the manuscript, subject to one minor suggestion. In the introduction the authors mention plastic waste due to covid - I expect that this actually represents a tiny proportion of the waste currently produced and suggest that the reference to covid is removed.

Reply to Reviewers' Comments

Comment from Reviewer #1:

Comment. The revision is well done but the claim that the Photothermal recycling is under mild condition should be reconsidered. The pressure is from 10bar to 40bar, which is not so mild.

Response: Thanks for the insightful suggestion. We have revised the title (Photothermal recycling of waste polyolefin plastics into liquid fuels with high selectivity under solvent-free conditions) in the revised manuscript.

Comment from Reviewer #2:

Comment. The extensive response to comments of the three reviewers and revisions are fine and the paper can be published.

Response: We thank the reviewer for the positive comment.

Comment from Reviewer #3:

Comment. The authors have extensively revised the manuscript and thoroughly addressed the key issues raised by the reviewers. I am pleased to accept the manuscript, subject to one minor suggestion. In the introduction the authors mention plastic waste due to covid - I expect that this actually represents a tiny proportion of the waste currently produced and suggest that the reference to covid is removed.

Response: Thanks for the useful suggestion. We have deleted the description in the revised manuscript.